# NatD promotes lung cancer progression by preventing histone H4 serine phosphorylation to activate Slug expression

Junyi Ju [1], Aiping Chen[2], Yexuan Deng[1], Ming Liu[1], Ying Wang[1], Yadong Wang[1], Min Nie[1], Chao Wang[3], Hong Ding[4], Bing Yao[1], Tao Gui[1], Xinyu Li[1], Zhen Xu[5], Chi Ma[1], Yong Song[2], Marc Kvansakul [6], Ke Zen[1], Chen-Yu Zhang[1], Cheng Luo [4], Ming Fang[3], David C.S. Huang[5], C.David Allis[7], Renxiang Tan[1,8], Changjiang Kathy Zeng[9], Jiwu Wei[2] & Quan Zhao[1]

N-α-acetyltransferase D (NatD) mediates N-α-terminal acetylation (Nt-acetylation) of histone H4 known to be involved in cell growth. Here we report that NatD promotes the migratory and invasive capabilities of lung cancer cells in vitro and in vivo. Depletion of NatD suppresses the epithelial-to-mesenchymal transition (EMT) of lung cancer cells by directly repressing the expression of transcription factor Slug, a key regulator of EMT. We found that Nt-acetylation of histone H4 antagonizes histone H4 serine 1 phosphorylation (H4S1ph), and that downregulation of Nt-acetylation of histone H4 facilitates CK2α binding to histone H4 in lung cancer cells, resulting in increased H4S1ph and epigenetic reprogramming to suppress Slug transcription to inhibit EMT. Importantly, NatD is commonly upregulated in primary human lung cancer tissues where its expression level correlates with Slug expression, enhanced invasiveness, and poor clinical outcomes. These findings indicate that NatD is a crucial epigenetic modulator of cell invasion during lung cancer progression.

[1] The State Key Laboratory of Pharmaceutical Biotechnology, School of Life Sciences, Nanjing University, Nanjing 210023, China. [2] Jiangsu Key Laboratory of Molecular Medicine, Medical School, Nanjing University, Nanjing 210093, China. [3] Institute of Life Sciences, Southeast University, Nanjing 210096, China. [4] Drug Discovery and Design Center, State Key Laboratory of Drug Research, Shanghai Institute of Materia Medica, Chinese Academy of Sciences, Shanghai 211203, China. [5] Department of Medical Biology, The Walter and Eliza Hall Institute of Medical Research, University of Melbourne, Melbourne, VIC 3052, Australia. [6] Department of Biochemistry, La Trobe University, Melbourne, VIC 3086, Australia. [7] Laboratory of Chromatin Biology and Epigenetics, The Rockefeller University, New York, NY 10065, USA. [8] State Key Laboratory Cultivation Base for TCM Quality and Efficacy, Nanjing University of Chinese Medicine, Nanjing 210046, China. [9] SQJ Biotechnologies Limited, Palo Alto, CA 94306, USA. Junyi Ju, Aiping Chen, and Yexuan Deng are contributed equally to this work. Correspondence and requests for materials should be addressed to C.K.Z. (email: cjzeng@sqjpharmaceuticals.com) or to J.W. (email: wjw@nju.edu.cn) or to Q.Z. (email: qzhao@nju.edu.cn)

N-α-terminal acetylation (Nt-acetylation) is one of the most common protein covalent modifications in eukaryotes, occurring in 80–90% of soluble proteins in humans and 50–70% in yeast[1–4]. This modification has a variety of biological roles, including regulation of protein degradation, protein–protein interactions, protein translocation, membrane attachment, apoptosis, and cellular metabolism[3, 5–7]. Nt-acetylation is catalyzed by N-α-acetyltransferases (NATs), which transfer the acetyl group from acetyl-coenzyme A (Ac-CoA) to the primary α-amino group of the N-terminal amino acid residue of a protein. In humans, six different NATs (NatA-NatF) have been identified to date based on their unique subunits and specific substrates[3]. NatD (also termed Nat4 or Patt1) mediates the Nt-acetylation of histone H4 and H2A exclusively, differentiating it from all other Nat family members, which target various substrates[8–10]. NatD contains only a single catalytic unit, Naa40p, and has no auxiliary subunit[3, 11].

NatD was originally identified in yeast, but the human NatD ortholog has also been characterized[11, 12]. In yeast, loss of NatD or its acetyltransferase activity produced a synthetic growth defect showing increased growth sensitivity to various chemicals including 3-aminotriazole, an inhibitor of transcription[13]. NatD was identified as a novel regulator of ribosomal DNA silencing during calorie restriction in yeast, which suggested that NatD might be critical for cell growth[14]. In line with this, male mice lacking NatD in liver showed decreased fat mass, and were protected from age-associated hepatic steatosis[15]. NatD is also linked to apoptosis of cancer cells. Intriguingly, in hepatocellular carcinoma, NatD was reported to enhance apoptosis, whereas in colorectal cells, depletion of NatD-induced apoptosis in a p53-independent manner[16, 17].

Epithelial-to-mesenchymal transition (EMT) is a key cellular program by which cancer cells lose their cell polarity and adhesion, and gain the migratory and invasive capabilities of mesenchymal cells, which is closely associated with metastasis[18]. Although this process was initially recognized during embryogenesis[18, 19], it has been extended to cancer cell stemness, drug resistance, and immunosuppression during cancer progression[20–22]. Recent studies have revealed interesting links between EMT and the control of the chromatin configuration resulting from histone modifications[23, 24]. However, the biological role of Nt-acetylation of histone by NatD during cancer progression involving EMT remains largely unknown.

In this study, we show that NatD-mediated N-α-terminal acetylation of histone H4 promotes lung cell invasion through antagonizing serine phosphorylation of histone H4 by CK2α The results demonstrate a critical interplay between transcriptional and epigenetic control during lung cancer progression associated with EMT of cancer cells, thus suggesting that NatD could be a potential therapeutic target for lung cancer.

## Results

**NatD expression associates with prognosis of lung cancer patients**. To investigate the clinical significance of NatD expression in patients with non-small cell lung cancer (NSCLC), we first examined *NatD* mRNA levels in human lung cancer tissues. Quantitative real-time PCR analysis showed that 69% (20/29) of lung cancer tissue samples showed significantly elevated *NatD* levels compared to adjacent normal tissue samples (Fig. 1a). We further examined expression of NatD by immunohistochemical staining (IHC) on two sets of human NSCLC tissue arrays containing 74 squamous carcinomas, 73 adenocarcinomas, and adjacent normal lung tissue controls (Supplementary Table 1). We found that NatD was significantly upregulated in both squamous carcinomas and adenocarcinomas compared with

normal lung tissues (Fig. 1b, c). Notably, NatD expression correlated with higher grade lymph node status (Fig. 1d). Importantly, the Kaplan–Meier survival analysis showed that lung cancer patients with high NatD expression had shorter overall survival (Fig. 1e). These results indicate that NatD expression levels are upregulated in human lung cancer tissues and correlate with poor prognosis in lung cancer, suggesting that NatD may promote cancer cell invasion during malignant progression.

**NatD is required for lung cancer cell migration and invasion in vitro**. To determine the effect of NatD on cell growth and mobility, we generated two independent, stable NatD knockdown human lung cancer H1299 cell lines (NatD-KD1 and NatD-KD2 cells) using lentiviral vectors containing different specific shRNAs targeting *NatD* mRNA. Because shRNA KD2 produced a somewhat better knockdown (Fig. 2a), unless both NatD-KD1 and NatD-KD2 cells are indicated, only NatD-KD2 cells were used. *NatD* mRNAs in NatD-KD1 and NatD-KD2 cells were reduced to 30% of *NatD* mRNAs in the scrambled control (Scr) cells determined by quantitative real-time PCR (Fig. 2a), and decreased protein levels of NatD were confirmed by western blot analysis (Fig. 2b). Correspondingly, levels of Nt-acetylation of histone H4 (Nt-ac-H4) were also significantly reduced in NatD knockdown cells compared with the Scr cells (Fig. 2b). We found that NatD knockdown cells grew at a similar rate as the Scr cells (Supplementary Fig. 1a), and no difference in numbers of apoptotic cells or in cell cycle was found between knockdown and Scr cells (Supplementary Fig. 1b, c). These results suggest that NatD has no effect on cell growth and survival of lung cancer cells. However, in a wound healing assay, NatD knockdown cells migrated significantly more slowly than Scr cells (Fig. 2c). Consistently, time-lapse cell-tracking analysis confirmed our observation dynamically, and showed lower random motility of NatD knockdown cells compared with the Scr cells (Fig. 2d). Furthermore, results from the transwell assay showed that cell migratory and invasive capabilities of lung cancer cells were significantly reduced in NatD knockdown cells compared with the Scr cells (Fig. 2e, f). Similar results were also obtained with another human lung cancer cell line, A549, when NatD was knocked down (Supplementary Fig. 2a–c). Thus, these results indicate that NatD is crucial for lung cancer cell migration and invasion in vitro.

**NatD promotes lung cancer cell invasion in vivo**. To further investigate the effect of NatD on lung cancer cell invasion in an in vivo model, luciferase-labeled Scr or NatD knockdown A549 cells were injected into severe combined immunodeficiency (SCID) mice via tail vein. Tumor growth was assessed by bioluminescent (BLI) imaging on days 1, 4, 7, 14, and 28. Mice receiving NatD knockdown cells exhibited significantly reduced lung cancer growth signals (photon radiance) compared with the mice receiving Scr cells (Fig. 3a). The effect of NatD knockdown was evident as early as day 4, suggesting that NatD expression was critical for extravasation and invasion of lung cancer cells even at an early stage (Fig. 3a). In turn, the colonization of cancer cells was also significantly inhibited, as we found that the number of tumor nodules in mice received NatD knockdown cells was decreased threefold relative to mice received Scr cells on day 28 (Fig. 3b). These findings were confirmed by quantitation of bioluminescence intensity in lungs (Fig. 3c).

In addition, we have generated a stable murine NatD knockdown Lewis lung carcinoma (LLC) cell line (Supplementary Fig. 3a). In vitro migratory and invasive capabilities of LLC cells were significantly decreased in the NatD knockdown LLC cells compared with the Scr cells (Supplementary Fig. 3b, c). Consistently, mice injected with NatD knockdown LLC cells via tail vein developed significantly fewer tumor nodules compared

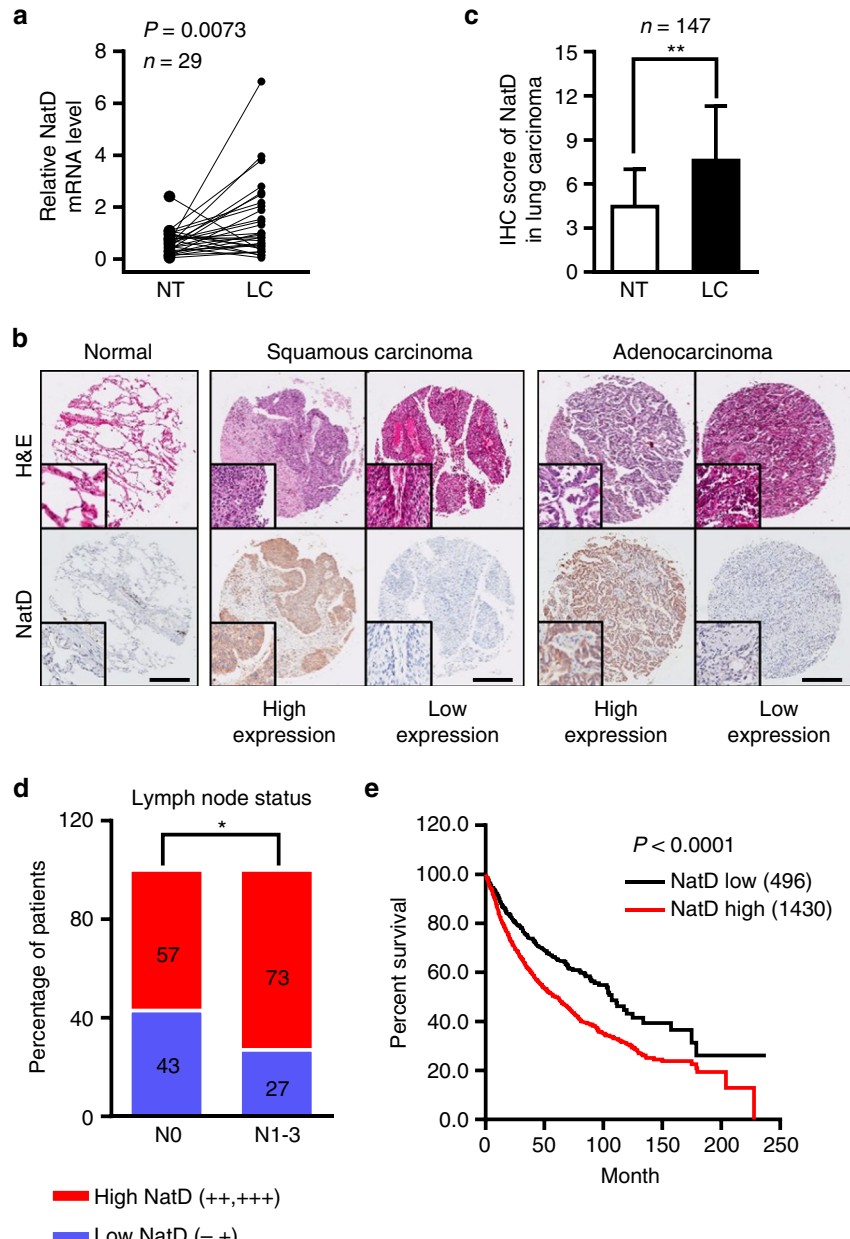

**Fig. 1** Upregulation of NatD in lung tissues correlates with enhanced invasiveness and poor prognosis of patients with lung cancer. **a** Quantitative real-time PCR analysis of *NatD* mRNA levels normalized to *GAPDH* in lung carcinoma (LC) and matched normal tissues (NT); $n = 29$, two-tailed Student's *t*-test; $P = 0.0073$ compared with matched normal tissue control. **b** Representative images of H&E staining and immunohistochemical (IHC) staining of NatD in matched normal tissues ($n = 147$), human lung squamous carcinoma ($n = 74$), and lung adenocarcinoma ($n = 73$) tissue samples. *Scale bars*, 500 μm. **c** Total IHC score of NatD in matched normal tissues (NT) and lung carcinoma (LC); mean ± s.d. of 147 pairs of tissue samples; two-tailed Student's *t*-test, \*\*$P < 0.01$ compared with matched normal tissue control. **d** Percentage of lung cancer patients with high expression and low expression of NatD stratified according to lymph node status (N0 or N1–3) ($n = 147$); two-sided Pearson $\chi^2$ test, \*$P < 0.05$. **e** Kaplan–Meier plots of overall survival of patients with lung cancer, stratified by NatD expression. Data were obtained from Kaplan–Meier plotter database[49]; log-rank test, $P < 0.0001$

with the Scr cells measured after 30 days' growth (Supplementary Fig. 3d), indicating that NatD knockdown markedly decreased the migratory and invasive ability of LLC cells. Consistently, in two orthotopic implantation models of lung cancers using human A549 and murine LLC cells, we found that the migration and invasion were significantly reduced in mice received NatD knockdown cells compared with the mice received Scr cells (Supplementary Fig. 9). These data indicate that the role of NatD is conserved between humans and mice, and that NatD has a critical role in promoting lung cancer cell invasiveness in vivo.

**Silencing NatD suppresses cancer cell EMT by downregulating Slug.** We next sought to determine how NatD controls the migratory and invasive phenotypes of cancer cells. In a TGF-β1-induced EMT experiment, we observed that Scr H1299 cells with an initial epithelial morphology developed a spindle-like appearance and mesenchymal morphology when treated with TGF-β1 (Fig. 4a). However, TGF-β1-treated NatD knockdown cells mostly retained their rounded epithelial morphology, and were largely, albeit incompletely, inhibited from undergoing EMT (Fig. 4a). This result suggests that NatD might be necessary for

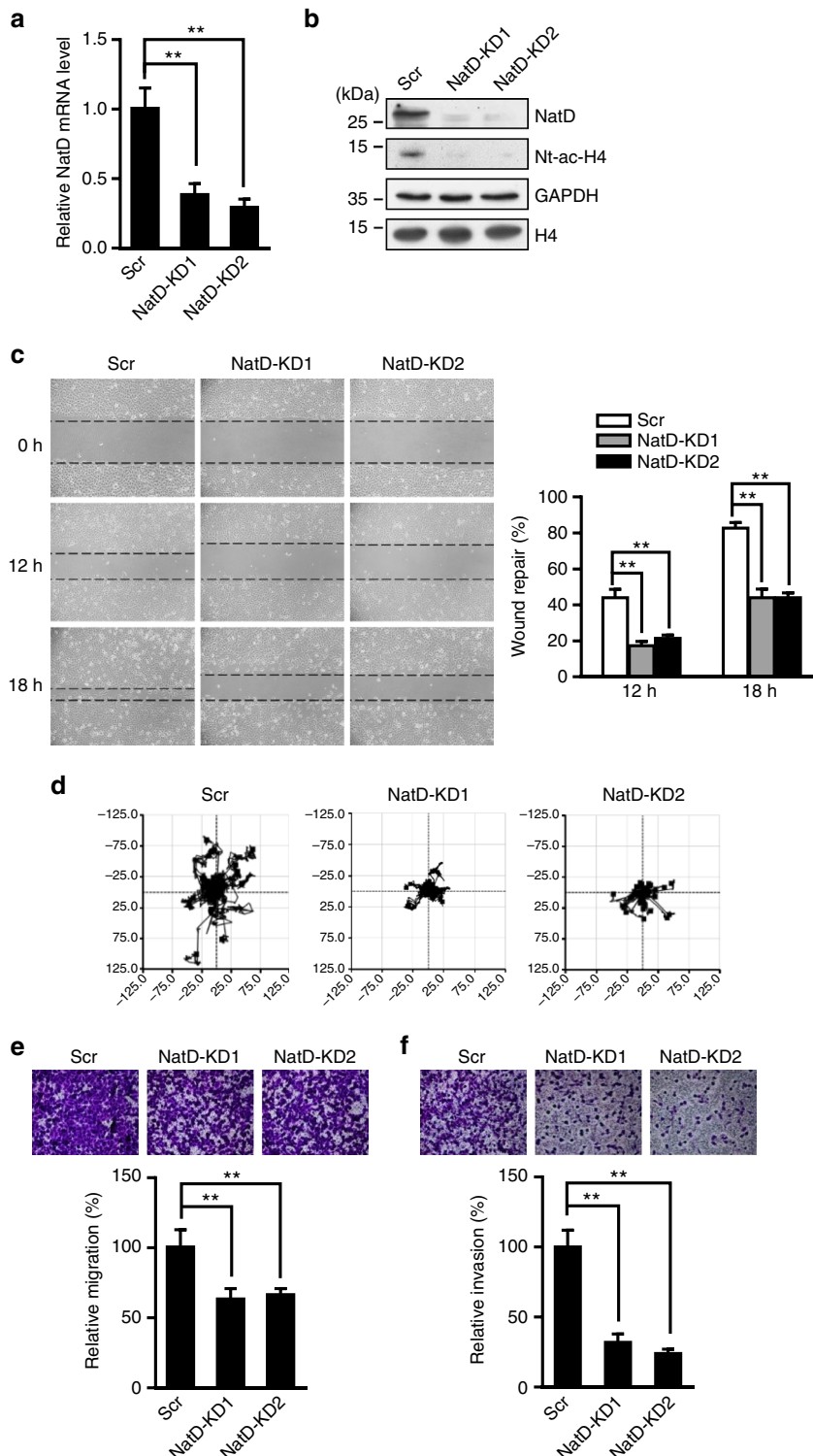

**Fig. 2** NatD is required for lung cancer cell migration and invasion in vitro. **a** Quantitative real-time PCR analysis of *NatD* mRNA levels normalized to *GAPDH* in scrambled control cells (Scr) and NatD-KD1 and NatD-KD2 cells. Results are shown as mean ± s.d. from three independent experiments. Two-tailed Student's *t*-test was used. **P < 0.01 compared to Scr control. **b** Western blot analysis of NatD and Nt-ac-H4 protein levels in scrambled, NatD-KD1, and NatD-KD2 cells. GAPDH and histone H4 served as loading controls. Blots are representative of three independent experiments. **c** Representative images from wound healing assay of scrambled, NatD-KD1, and NatD-KD2 cells from three independent experiments (*left panels*). Wound healing assay results are quantified in the histogram (*right panel*). Results are shown as mean ± s.d. from three independent experiments. Two-tailed Student's *t*-test was used. **P < 0.01 compared to Scr control. **d** Representative images of the migration of scrambled, NatD-KD1, and NatD-KD2 cells in a time-lapse cell tracker migration assay from three independent experiments. Representative images of the migration (**e**) and invasion (**f**) of scramble, NatD-KD1, and NatD-KD2 cells with transwell assay from three independent experiments (*top panel*). Cell counts for the corresponding assays of at least four random microscope fields (×100 magnification). Cell migration and invasion are expressed as a percentage of control (*bottom panel*). Results are shown as mean ± s.d. from three independent experiments. Two-tailed Student's *t*-test was used. **P < 0.01 compared to Scr control

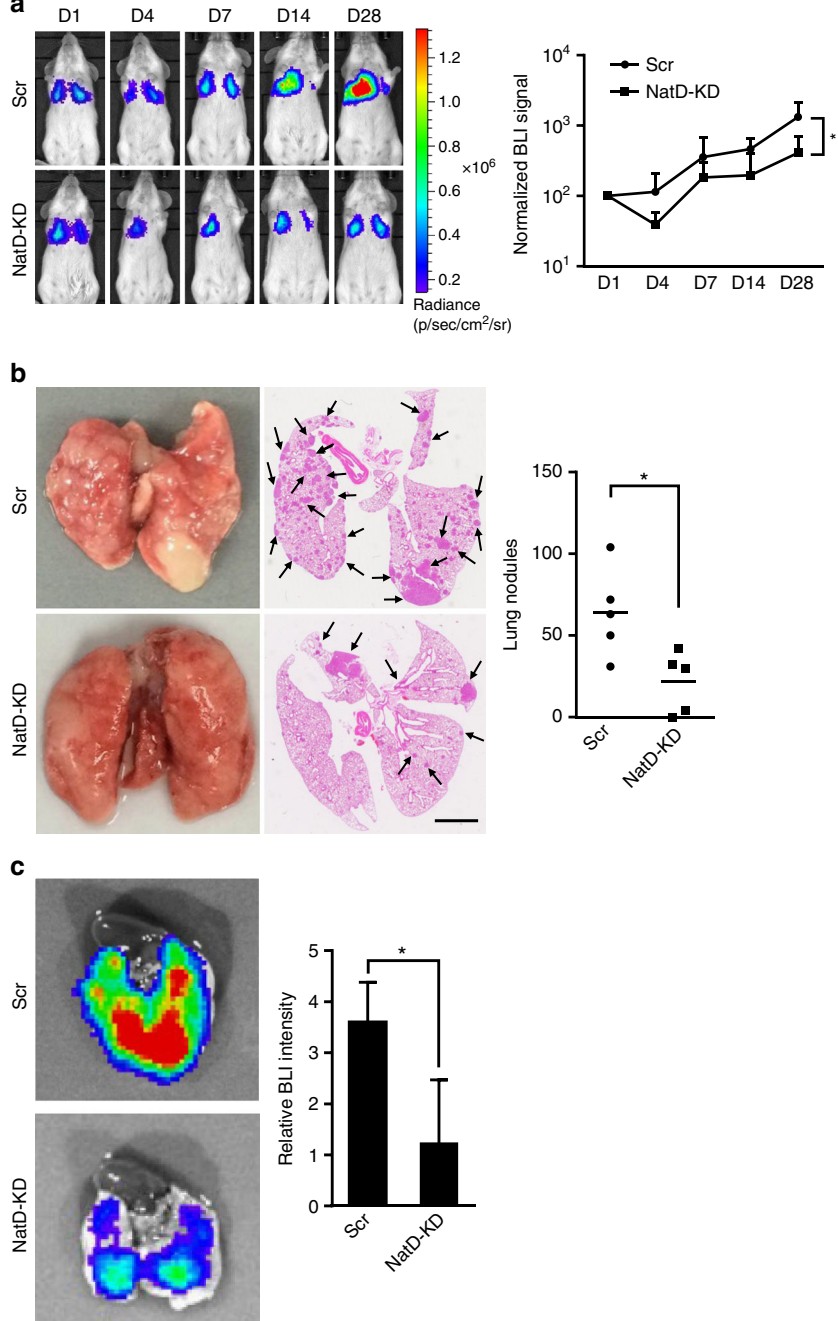

**Fig. 3** NatD promotes lung cancer cell invasion in a xenograft mouse model. **a** (*left*) Representative bioluminescent (BLI) images acquired at the indicated time points after intravenous injection of SCID mice with Scr or NatD-KD A549 cells. NatD-KD2 cells were used because the KD2 shRNA produced a better knockdown effect. Pseudocolor heat-maps indicate intensity of bioluminescence from low (*blue*) to high (*red*) (D, day). (*right*) Normalized BLI signals of lung tumors of corresponding mice ($n = 5$ for each group) recorded at the indicated time points. Results are shown as mean ± s.d. from five mice. Two-tailed Student's $t$-test was used. *$P < 0.05$ compared to Scr control. **b** (*left*) Representative images of lung nodules of SCID mice acquired 28 days after intravenous injection with Scr or NatD-KD A549 cells. (*middle*) Representative images of H&E stained histological sections of lungs from SCID mice. *Scale bars*, 2 mm. Arrows indicate major metastatic nodules. (*right*) Box plot showing numbers of lung nodules from corresponding mice ($n = 5$ for each group). Results are shown as mean ± s.d. from five mice. Two-tailed Student's $t$-test was used. *$P < 0.05$ compared to Scr control. **c** (*left*) Representative images showing luciferase activity in lungs from SCID mice as in (**b**). (*right*) Quantification of total lung bioluminescence from SCID mice as in (**b**) ($n = 5$ for each group). Results are shown as mean ± s.d. from five mice. Two-tailed Student's $t$-test was used. *$P < 0.05$ compared to Scr control

EMT. Loss of component molecules of cell adhesion and tight junctions is the hallmark of EMT in cancer[25, 26]. We then examined changes in expression levels of key EMT-related transcription factors and markers in lung cancer cells after NatD knockdown under basal conditions in the absence of TGF-β1.

Quantitative real-time PCR showed that NatD knockdown increased the expression of the epithelial marker *E-cadherin*, but reduced the expression of mesenchymal markers, *N-cadherin*, and *Vimentin* (Fig. 4b). Interestingly, in terms of transcription factors, only the expression of *Slug* was significantly repressed in NatD

knockdown cells, whereas the expression of *Twist1*, *Snail*, *Zeb1*, or *Zeb2* was not changed in this context (Fig. 4b; Supplementary Fig. 5a). The protein levels of E-cadherin, N-cadherin, Vimentin, and Slug were also altered consistently as determined by western blot analysis (Fig. 4c). Immunofluorescence staining experiments further confirmed that E-cadherin staining was significantly increased and N-cadherin was decreased in cell-to-cell junctions in NatD knockdown cells compared with the Scr cells (Fig. 4d).

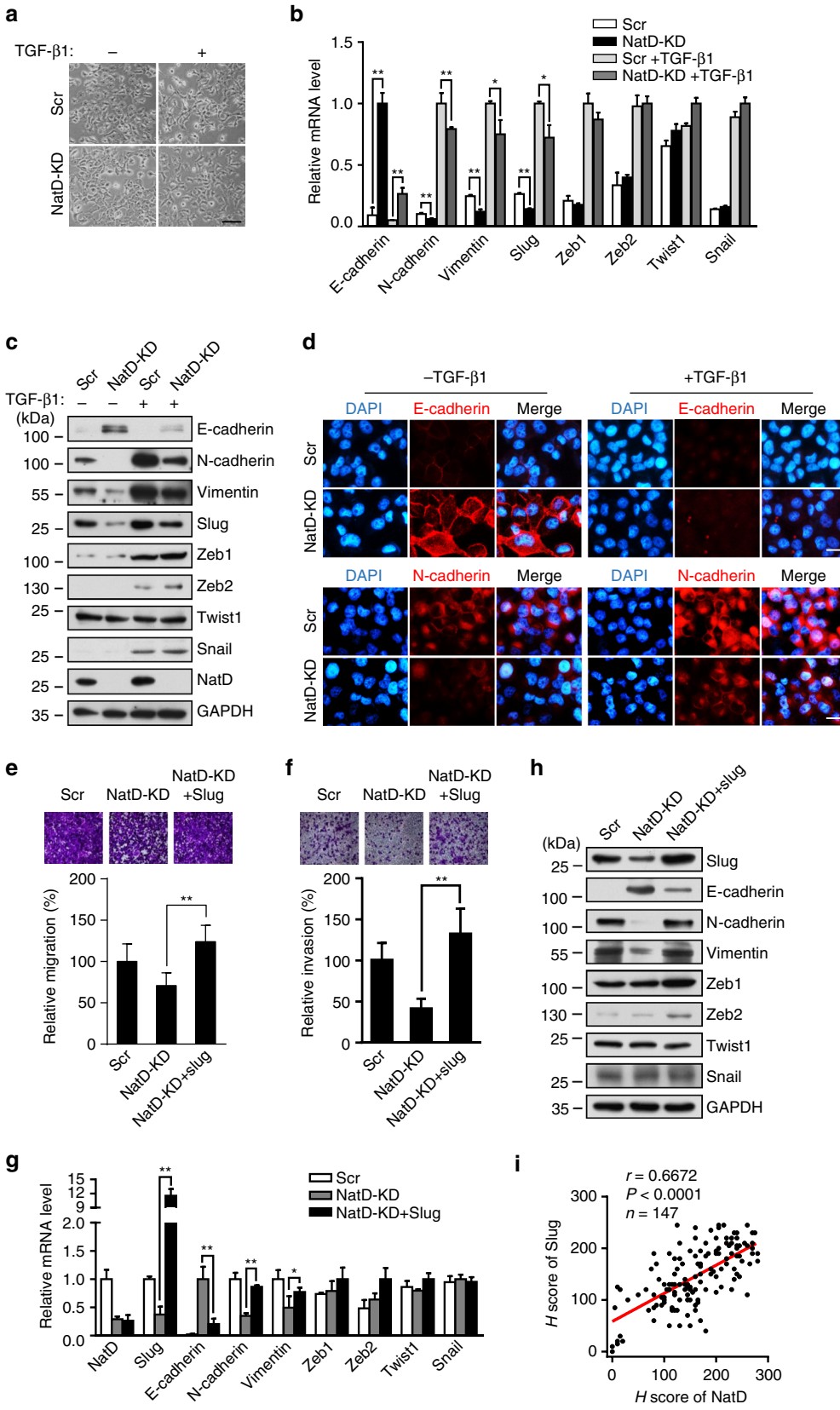

We further found that NatD knockdown blocked changes of expression levels of EMT marker genes E-cadherin, N-cadherin, Vimentin, and transcription factor Slug in the presence of TGF-β1 relative to basal condition (Fig. 4b–d). In addition, we analyzed the expression of a spectrum of key proliferation-related and cell cycle-related genes, including *CCND1*, *p21*, *p27*, *p57*, *p16*, *p18*, *p19*, *CHEK2*, *E2F1*, *CCND2*, *KRAS*, *PTEN*, *c-Myc*, and *PCNA* in H1299 cells by quantitative real-time PCR[27–29]. We found that NatD knockdown did not affect the expression of these genes except for *CCND2*, a gene which may function in cell migration as well (Supplementary Fig. 4a)[30]. Taken together, these data indicate that NatD is mainly required for maintaining the mesenchymal phenotype, and its downregulation inhibits EMT of lung cancer cells. Consistent results were also obtained in murine LLC cells and human A549 cells; NatD knockdown increased the expression of E-cadherin and decreased the expression of Slug, N-cadherin, and Vimentin in LLC cells (Supplementary Fig. 3a, e, f) and in A549 cells (Supplementary Fig. 4b–d).

Slug is a critical transcriptional regulator of EMT that suppresses E-cadherin expression by direct binding to the *CDH1* promoter[31]. Thus, we tested the possibility that enforced expression of Slug would compensate for NatD knockdown. As expected, migratory and invasive capabilities of NatD knockdown H1299 cells were restored by ectopic expression of Slug (Fig. 4e, f), which was accompanied by suppression of E-cadherin and increased expression of N-cadherin and Vimentin (Fig. 4g, h). Similar results were also obtained in NatD knockdown A549 cells in which Slug expression was ectopically enforced (Supplementary Fig. 4e–h). These results suggest that the ability of NatD to promote EMT likely involves activation of Slug expression.

To probe Slug expression in patients with NSCLC, we performed IHC staining on the same set of human NSCLC tissue arrays containing 74 squamous carcinomas, 73 adenocarcinomas, and adjacent normal lung tissue controls (Supplementary Table 1) with anti-Slug antibody. We found that Slug expression was also significantly upregulated in both squamous carcinomas and adenocarcinomas compared with the normal lung tissues (Supplementary Fig. 5b, c). More interestingly, the expression of Slug and NatD correlated well across all NSCLC samples analyzed (Fig. 4i). This is supported by the Kaplan–Meier survival analysis showing that lung cancer patients with high Slug expression had shorter overall survival (Supplementary Fig. 5d). These results further suggest that NatD may positively regulate Slug expression to promote cancer cell invasion during lung cancer progression.

**Regulation of Slug by NatD is acetyltransferase activity-dependent**. NatD is an N-α-terminal acetyltransferase that exclusively modifies histone H4 and H2A. To determine whether the regulation of Slug expression and EMT by NatD was acetyltransferase activity-dependent, we constructed a mutant form of NatD (NatDΔ) in which four amino acids (RRKG, aa147–150) located in the acetyl-CoA (Ac-CoA)-binding motif were deleted. Of note, the Ac-CoA binding motif is highly conserved from yeast to humans[11]. The loss of acetyltransferase activity of NatDΔ was confirmed in an in vitro acetylation assay of a histone H4 N-terminal peptide using $^3$H-Ac-CoA as an acetyl donor (Fig. 5a). Nt-ac-H4 was also assessed by western blot analysis with an anti-Nt-ac-H4 antibody (Fig. 5b; Supplementary Fig. 6a, b). Moreover, wild-type NatD, but not NatDΔ, was able to mediate Nt-acetylation of histone H4 in histones extracted from H1299 cells (Fig. 5c). Nt-acetylation on histone H2A was not mediated by either NatDΔ or NatD in this context (Fig. 5c). Consistently, we detected significantly reduced expression levels of Slug as well as N-cadherin and Vimentin, and increased expression levels of E-cadherin in NatDΔ cells compared with the wild-type NatD cells by both quantitative RT-PCR and western blot analysis (Fig. 5d, e). Furthermore, the transwell assay showed that cell migratory and invasive capabilities of lung cancer cells were significantly reduced in NatDΔ cells compared with the wild-type NatD cells (Fig. 5f, g). These results indicate that Nt-acetylation of histone H4 by NatD is critical for maintaining the expression of Slug in lung cancer cells.

**Nt-acetylation of histone H4 antagonizes phosphorylation of histone H4 serine 1 to regulate Slug expression**. In eukaryotic cells, NatD catalyzes Nt-ac-H4, which occurs on the first serine residue of histone H4 (H4S1). However, this serine residue can also be phosphorylated (H4S1ph) by casein kinase 2α (CK2α)[32, 33]. In NatD knockdown cells, compared with the Scr cells, as expected, levels of Nt-ac-H4 were greatly decreased (Fig. 6a). Interestingly, we found that the levels of the histone mark H4S1ph were significantly increased in NatD-KD cells compared with the Scr cells in western blot analysis of total cellular lysates (Fig. 6a). In addition to H4S1ph, we also observed that histone mark H4R3me2a was slightly increased, whereas H4R3me2s was slightly decreased in NatD-KD cells compared with the Scr cells (Fig. 6a). No significant change in levels of histone H4K5ac, H4K8ac, or H4K12ac was found between NatD-KD cells and Scr cells (Fig. 6a).

We found that enrichment of Nt-ac-H4 was significantly reduced on the Slug promoter in NatD-KD cells compared with the Scr cells (Fig. 6b). Consistent with the pan-cellular western blot analysis, we observed significantly increased enrichment levels of H4S1ph on the Slug promoter in NatD-KD cells compared with the Scr cells (Fig. 6b). However, in contrast to pan-cellular levels of the histone marks, we found significantly

---

**Fig. 4** Silencing NatD suppresses cancer cell EMT by downregulating Slug. **a** Representative phase contrast images of Scr and NatD-KD H1299 cells treated with TGF-β1. Data are representative of three independent experiments. Scale bar, 100 μm. **b** Quantitative real-time PCR analysis of mRNA levels of indicated key EMT-related genes in Scr and NatD-KD H1299 cells normalized to *GAPDH* in the absence or presence of TGF-β1. Results are shown as mean ± s.d. of three independent experiments. Two-tailed Student's *t*-test was used. \*\**P* < 0.01 or \**P* < 0.05 compared with the indicated control. **c** Western blot analysis of indicated protein levels in Scr and NatD-KD H1299 cells in the absence or presence of TGF-β1. GAPDH served as a loading control. Data are representative of three independent experiments. **d** Immunofluorescence analysis of Scr and NatD-KD H1299 cells in the absence or presence of TGF-β1 stained for E-cadherin and N-cadherin. Data are representative of three independent experiments. *Scale bar*, 20 μm. Migration (**e**, *top*) and invasion (**f**, *top*) of Scr cells, NatD-KD cells, and NatD-KD cells with enforced Slug expression (NatD-KD + Slug). (*bottom panels*) Cells were counted in at least four random microscope fields (×100 magnification) for the corresponding assays; migration and invasion are expressed as a percentage of control. Results are shown as mean ± s.d. of three independent experiments. Two-tailed Student's *t*-test was used. \*\**P* < 0.01 compared with the indicated control. **g** Quantitative real-time PCR analysis of the mRNA levels of *NatD* and indicated key EMT-related genes (normalized to *GAPDH*) in Scr cells, NatD-KD cells, and NatD-KD + Slug cells. Results are shown as mean ± s.d. of three independent experiments. Two-tailed Student's *t*-test was used. \*\**P* < 0.01 or \**P* < 0.05 compared with Scr or indicated control. **h** Western blot analysis of indicated protein levels in Scr cells, NatD-KD cells, and NatD-KD + Slug cells. GAPDH served as a loading control. Data are representative of three independent blots. **i** Pearson correlation scatter plot of the H score of Slug and NatD in human lung carcinoma (*n* = 147); *r* = 0.6672, *P* < 0.0001

reduced enrichment levels of H4R3me2a and H4K5ac on the Slug promoter in NatD-KD cells compared with the Scr cells (Fig. 6b). Enrichment levels of H4R3me2s on the Slug promoter were unchanged in NatD-KD cells compared with the Scr cells (Fig. 6b). We also found significantly reduced enrichment levels

of H3K4me3 and increased enrichment levels H3K27me3 in NatD-KD cells compared with the Scr cells (Fig. 6c). Of note, these changes in histone marks were consistent with down-regulation of Slug expression by NatD knockdown. Enrichment levels of H4S1ph, H3K4me3, and H3K27me3 on the promoters of *Zeb1*, *Zeb2*, *Twist1*, and *Snail* were unchanged although those of Nt-ac-H4 were reduced in NatD-KD cells compared with the Scr cells (Supplementary Fig. 7a–d). Importantly, an antagonistic relationship between Nt-ac-H4 and H4S1ph was dependent on the acetyltransferase activity of NatD (Fig. 6d, e). These results suggest that Nt-acetylation and phosphorylation of histone H4S1 are antagonistic, and histone mark Nt-ac-H4 can communicate with other histone modifications to co-ordinately modulate Slug gene expression.

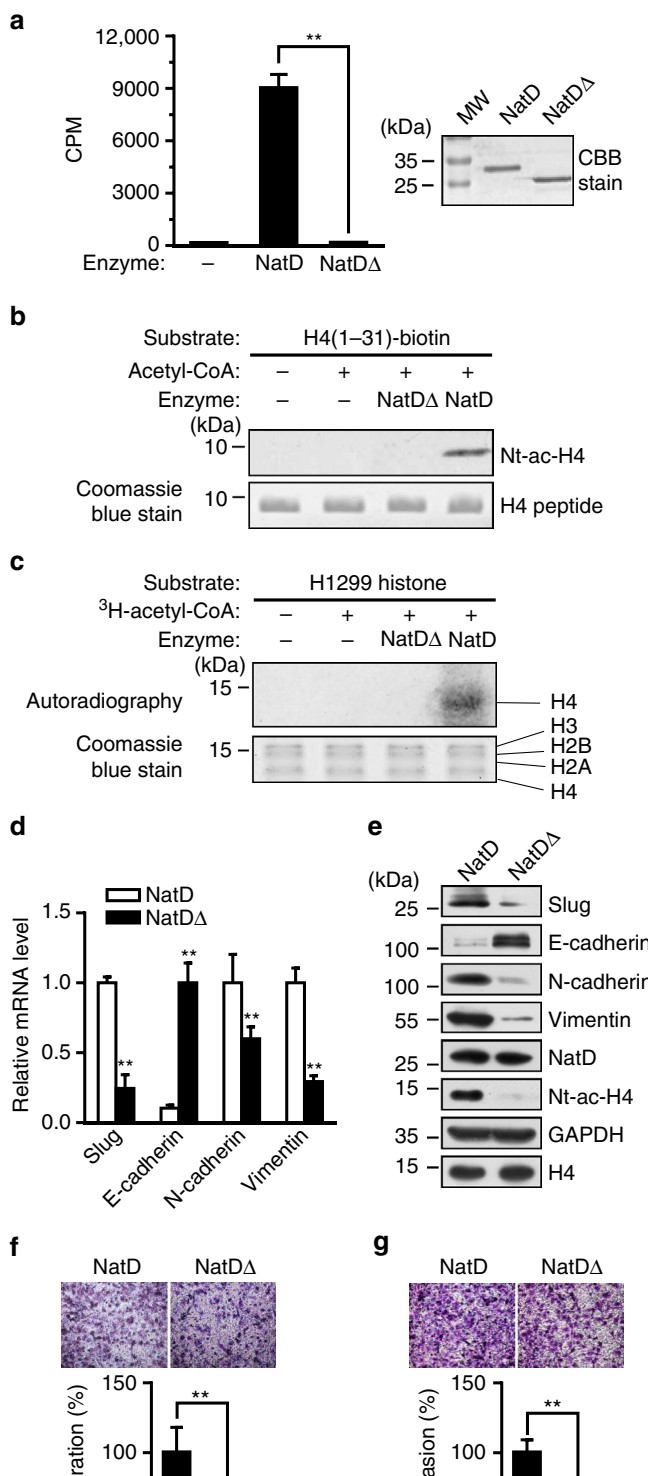

**Downregulation of Nt-acetylation of histone H4 facilitates binding of CK2α to histone H4 in lung cancer cells**. We have shown that levels of histone marker H4S1ph were significantly increased when NatD was knocked down in lung cancer cells. Therefore, we wanted to determine whether CK2α, a catalytic subunit of CK2 responsible for triggering phosphorylation of histone H4S1[32, 33], was upregulated due to NatD knockdown. Quantitative RT-PCR detection and western blot analysis showed no increase in the levels of either *CK2α* mRNA or protein in NatD knockdown cells compared with the Scr cells (Supplementary Fig. 8a, b). These results indicated that the increased levels of H4S1ph in NatD knockdown cells were not due to elevated expression of CK2α. Thus, we suspected that the increased levels of H4S1ph in NatD knockdown cells were because more CK2α was being shuttled into the nucleus after NatD knockdown. To test this possibility, we performed a confocal immuno-fluorescence assay using specific anti-CK2α antibody in both NatD knockdown cells and Scr cells. We found that nearly 100% of CK2α in NatD knockdown cells was localized in the nucleus,

**Fig. 5** Regulation of Slug by NatD is acetyltransferase activity-dependent. **a** (*left*) In vitro acetylation assay showing the catalytic activity of NatDΔ and wild-type NatD (CPM, counts per minute). Data are mean ± s.d. of three independent experiments; Student's *t*-test, **$P < 0.01$ compared with wild-type NatD. (*right*) SDS-PAGE analysis of purified recombinant NatDΔ and wild-type NatD proteins from *E. coli* stained by Coomassie brilliant blue (CBB). MW, protein molecular weight markers. **b** (*top*) Western blot analysis of an H4 (1–31) peptide from in vitro acetylation assay in the presence of NatDΔ or wild-type NatD. (*bottom*) H4 (1–31) peptide shown by Coomassie blue staining. Blots are representative of three independent experiments. **c** (*top*) Autoradiographic image showing products from in vitro acetylation assay using histones as substrates extracted from H1299 cells. Results are representative of three independent experiments. (*bottom*) Histones shown by Coomassie blue staining. **d** Quantitative real-time PCR analysis of mRNA levels of *Slug*, *E-cadherin*, *N-cadherin*, and *Vimentin* normalized to *GAPDH* in H1299 cells overexpressing NatDΔ or wild-type NatD. Data are mean ± s.d. of three independent experiments; Student's *t*-test, **$P < 0.01$ compared with the wild-type NatD. **e** Western blot analysis of indicated proteins from H1299 cells overexpressing NatDΔ or wild-type NatD. GAPDH and histone H4 served as loading controls. Data are representative of three independent experiments. **f**, **g** Representative images of the migration (**e**) and invasion (**f**) of H1299 cells overexpressing NatDΔ or wild-type NatD with transwell assay from three independent experiments (*top panel*). Cell counts for the corresponding assays of at least four random microscope fields (×100 magnification). Cell migration and invasion are expressed as a percentage of control (*bottom panel*). Results are shown as mean ± s.d. from three independent experiments. Two-tailed Student's *t*-test was used. **$P < 0.01$ compared with the indicated control

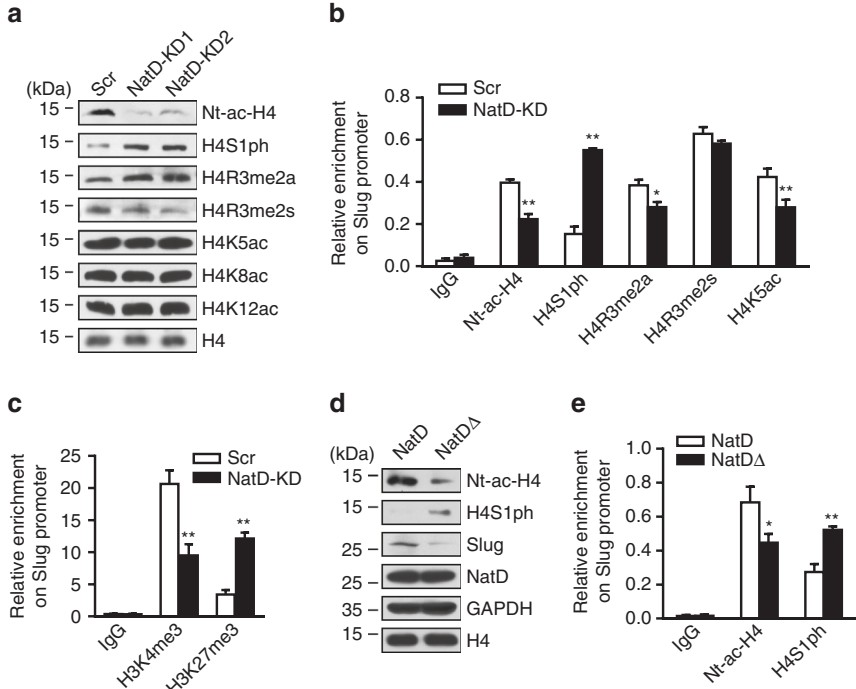

**Fig. 6** Nt-acetylation of histone H4 antagonizes phosphorylation of histone H4 serine 1 to regulate Slug expression. **a** Western blot analysis of indicated histone H4 modifications in Scr and NatD-KD H1299 cells. Histone H4 served as a loading control. **b** ChIP analysis of the enrichment of indicated histone H4 modifications on the Slug promoter in Scr and NatD-KD H1299 cells. IgG served as a negative control. Results are shown as mean ± s.d. from three independent experiments; two-tailed Student's *t*-test, *$P < 0.05$, **$P < 0.01$ compared with the Scr control. **c** ChIP analysis of the enrichment of H3K4me3 and H3K27me3 on the Slug promoter in Scr and NatD-KD H1299 cells. IgG served as a negative control. Results are shown as mean ± s.d. from three independent experiments; two-tailed Student's *t*-test, **$P < 0.01$ compared with the Scr control. **d** Western blot analysis of indicated proteins from H1299 cells overexpressing NatDΔ or wild-type NatD. Histone H4 served as a loading control. **e** ChIP analysis of the enrichment of Nt-ac-H4 and H4S1ph on Slug promoter in H1299 cells overexpressing NatDΔ or wild-type NatD. IgG served as a negative control. Results are shown as mean ± s.d. from three independent experiments; two-tailed Student's *t*-test, *$P < 0.05$, **$P < 0.01$ compared with the wild-type control

but in Scr cells only about 70% of CK2α was localized in the nucleus (Fig. 7a). There were no detectable levels of CK2α in the cytoplasm of NatD knockdown cells on western blots, consistent with the immunofluorescence assay (Fig. 7a, b). In contrast, in Scr cells, expression of CK2α was also detected in the cytoplasm as well as in the nucleus (Fig. 7a, b). These results provide evidence indicating that NatD knockdown resulted in re-localization of CK2α to the nucleus. This finding raised the question of what is the consequence of the movement of CK2α from the cytoplasm to the nucleus.

The observation that Nt-ac-H4 and H4S1ph are antagonistic, and that NatD knockdown results in additional shuttling of CK2α into the nucleus leading to significantly increased phosphorylation of H4S1, suggests that, in NatD-replete cells, Nt-acetylation of histone H4 may obstruct binding of CK2α to histone H4 preventing phosphorylation. To examine this possibility, we performed a peptide pulldown assay using C-terminal biotin-tagged 31 amino acid N-terminal peptides of histone H4 in which the Ser1 residue was either acetylated (pNt-ac-H4) or non-acetylated (pH4), or mutated to alanine (pH4S1A), or using C-terminal biotin-tagged 20 amino acid N-terminal peptides of histone H3 without N-terminal acetylation (pH3). We analyzed the eluates from pulldowns by western blot with an antibody against CK2α. We found significant binding of CK2α to non-acetylated H4 peptide but not to Nt-ac-H4 peptide, H4S1A peptide, or H3 peptide (Fig. 7c). We determined that non-acetylated H4 peptide was directly bound by CK2α by microscale thermophoresis (MST) assay using purified recombinant CK2α (amino acids 1–335)[34] expressed in *E. coli* (Fig. 7d, e). The data fit

a one-site-binding model with $K_D$ values of 33.5 ± 2.87 μM for CK2α binding to non-acetylated H4 peptide (Fig. 7e). No binding of CK2α to Nt-acetylated H4 peptide was detected (Fig. 7e). Consistent with these results, the enrichment of CK2α on the Slug promoter was significantly increased in NatD-KD cells compared with the Scr cells (Fig. 7f). Indeed, knockdown of CK2α by RNA interference significantly increased Slug expression, particularly in NatD-KD cells (Fig. 7g–h). Thus, downregulation of Nt-acetylation of histone H4 facilitated nuclear accumulation of CK2α and it's binding to histone H4 in lung cancer cells, resulting in increased phosphorylation of histone H4 serine 1. These results demonstrate that NatD-mediated N-α-terminal acetylation of histone H4 prevents serine 1 phosphorylation of histone H4 by blocking the binding of CK2α to histone H4.

## Discussion

Histone modification has an essential role in gene regulation. However, the function of N-α-terminal acetylation of histone H4 has remained uncertain even though this modification is abundant, and the corresponding enzyme NatD is highly conserved in eukaryotes[3, 4]. In this study, we show that NatD-mediated Nt-acetylation of histone H4 antagonizes serine phosphorylation to promote EMT in lung cancer. This process is depicted in the model shown in Fig. 8. High NatD expression in lung cancer samples was correlated with high Slug expression, enhanced invasiveness, and reduced patient survival. These findings suggest that NatD is a key epigenetic regulator of cell invasion during lung cancer progression.

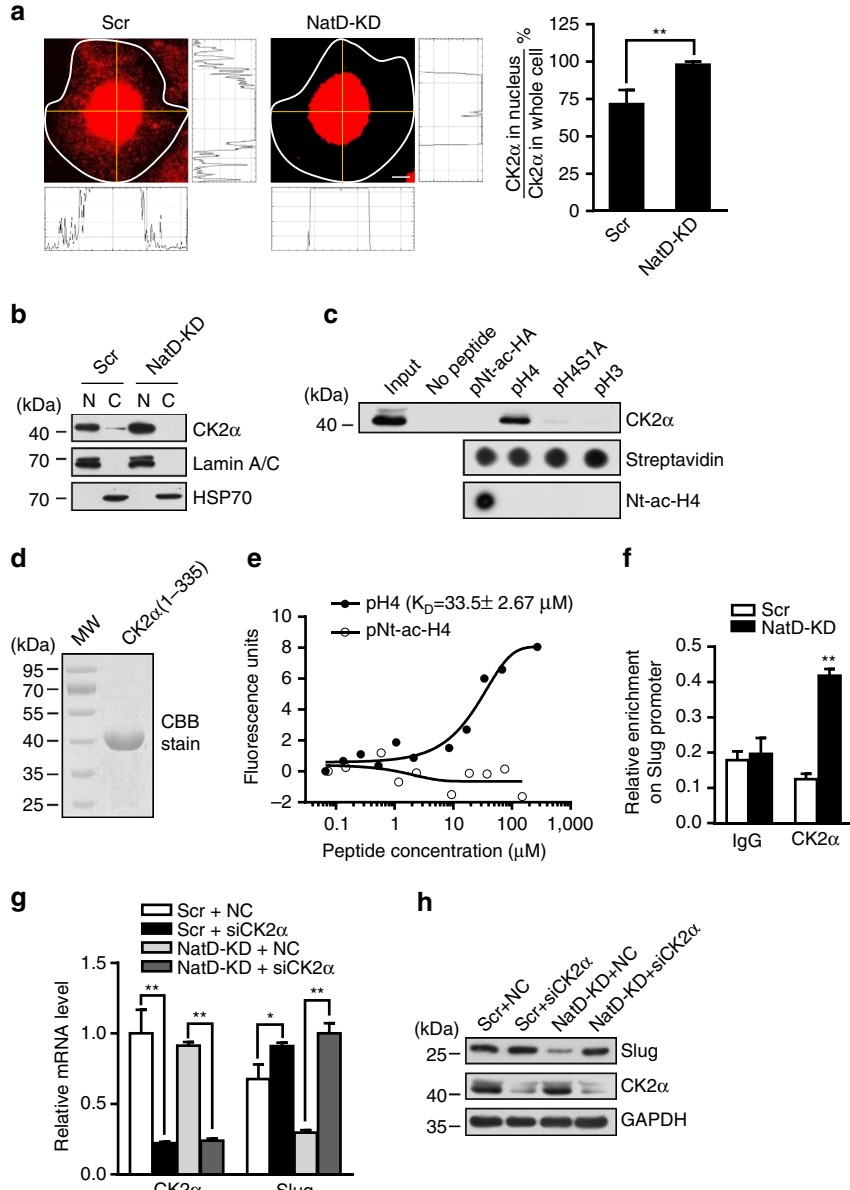

**Fig. 7** Downregulation of Nt-acetylation of histone H4 facilitates nuclear accumulation of CK2α and its binding to histone H4 in lung cancer cells. **a** Representative confocal scanning images of CK2α localization in Scr and H1299 cells (*left panel*). The staining intensity of CK2α was quantified by software ImageJ from NIH (*right diagram*). *Scale bars*, 5 μm. Results are shown as mean ± s.d. from more than 30 cells from three independent experiments; two-tailed Student's *t*-test, **$P < 0.01$ compared with the Scr control. **b** Western blot analysis of CK2α distribution in scrambled and NatD-KD H1299 cells with indicated antibodies. N nucleus, C cytoplasm. **c** Peptide pulldown assay to detect the interaction between H4 (1–31) or H3 (1–20) peptide (pNt-ac-H4, pH4, pH4S1A, or pH3) and CK2α in H1299 cell nuclear extracts (*top panel*). Equal peptide biotinylated on C terminus is shown by dot blot analysis with streptavidin (*middle panel*). Nt-ac-H4 was confirmed by dot blot analysis with anti-Nt-ac-H4 antibody (*bottom panel*). **d** SDS-PAGE analysis of purified recombinant CK2α (1–335) from *E. coli* stained by Coomassie brilliant blue (CBB). MW: protein molecular weight markers. **e** MST assay to identify direct interactions between CK2α (1–335) and H4 or Nt-ac-H4 peptide. The dissociation constant ($K_D$) between CK2α (1–335) and H4 peptide is 33.5 ± 2.67 μM. **f** ChIP analysis of the enrichment of CK2α on the *Slug* promoter in Scr and NatD-KD H1299 cells. IgG served as a negative control. Results are shown as mean ± s.d. from three independent experiments; two-tailed Student's *t*-test, **$P < 0.01$ compared with the indicated control. **g** Quantitative real-time PCR analysis of mRNA levels of *CK2α* and *Slug* normalized to *GAPDH* in Scr and NatD-KD H1299 cells in the absence or presence of siRNA to CK2α. NC, siRNA mimics negative control. Data are mean ± s.d. of three independent experiments; Student's *t*-test, *$P < 0.05$, **$P < 0.01$ compared with the indicated control. **h** Western blot analysis of indicated proteins from Scr and NatD-KD H1299 cells in the absence or presence of siRNA to CK2α. GAPDH served as a loading control

A large body of evidence suggests that EMT is an important driver of cancer progression[20–22]. Histone modifications have been shown to link closely to EMT[23, 24]. To undergo EMT, cancer cells need to acquire epigenetic changes other than genetic changes[23, 24, 35]. This study demonstrated that NatD can trigger Nt-acetylation of histone H4 on the *Slug* promoter to promote

EMT of lung cancer cells. These findings identify a new function for NatD in gene expression regulation, and extend our understanding of epigenetic regulation of EMT via Nt-acetylation of histone H4. Slug, a key regulator of EMT, has been identified as one of the major drivers of chemoresistance, and is associated with cancer stem cell properties[36, 37]. In our results, we found that

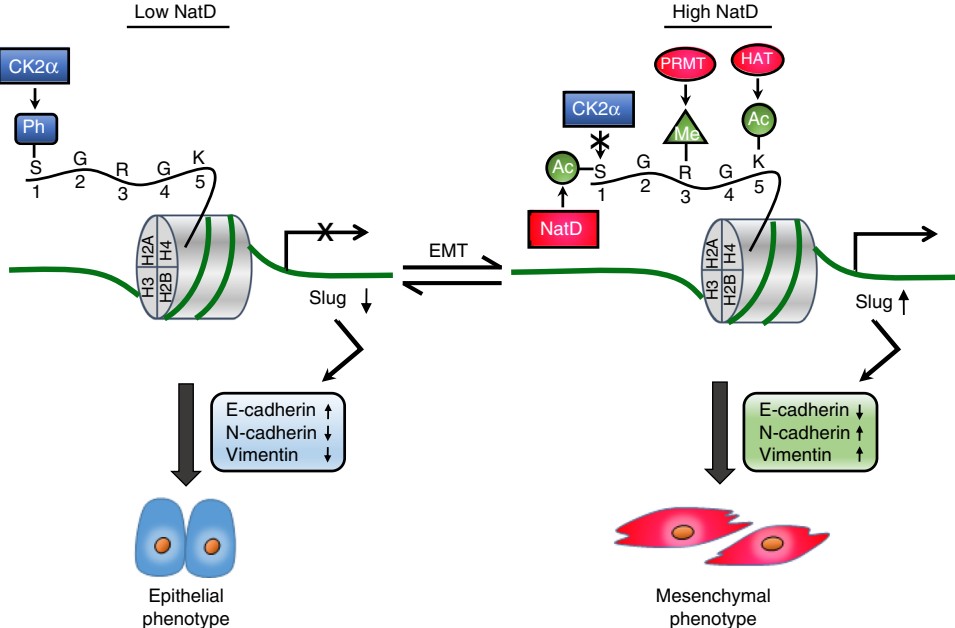

**Fig. 8** Model for the role of NatD in EMT. In human lung cancer cells, when NatD levels are low, CK2α phosphorylates histone H4S1, which silences Slug expression, retaining tumor cells in the epithelial state. When NatD levels are high, NatD acetylates histone H4 serine 1 (S1), which blocks CK2α-mediated S1 phosphorylation, and allows arginine 3 (R3) methylation and lysine 5 (K5) acetylation to activate Slug gene expression, promoting EMT to generate mesenchymal tumor cells, which increases cancer cell invasion and metastasis. PRMT protein arginine methyltransferase, HAT histone acetyltransferase

Slug is a direct epigenetic target of NatD to mediate EMT processes of lung cancer cells. More importantly, the expression levels of Slug correlate intimately with those of NatD in lung cancer tissues. These data suggest that NatD may also be linked to chemoresistance and cancer cell stemness which deserve further investigation in the future. We favor the hypothesis that Slug is the key regulator of EMT. However, given the capacity of NatD to regulate expression of multiple genes, we cannot at this point completely rule out the possibility that other genes directly regulated by NatD might also contribute to migration and invasion independent of Slug expression.

Interestingly, we observed that Nt-acetylation by NatD and phosphorylation of histone H4S1 by CK2α were antagonistic on the Slug promoter. Our results demonstrated that removal of Nt-acetylation facilitated nuclear accumulation of CK2α and its binding to histone H4 in lung cancer cells resulting in phosphorylation of histone H4 Ser1 by CK2α, suggesting that Nt-acetylation of histone H4 obstructs binding of CK2α. These data unveil a mechanistic switch from Nt-acetylation of histone H4 by NatD to phosphorylation of histone H4 Ser1 by CK2α, although the reason why NatD knockdown led to cytoplasm-to-nucleus shuttling of CK2α is currently unclear.

In addition to increased enrichment of H4S1ph, reduced Nt-acetylation of H4 also resulted in decreased enrichment of H4K5ac and H4R3me2a on the Slug promoter. Histone H4S1ph has been shown to have a temporal inverse relationship with H4K5/K8/K12ac during yeast sporulation and mammalian spermatogenesis[38], and is inhibitory to acetylation on histone H4K5/K8/K12ac during DNA damage[39]. Thus, our results indicate that H4S1ph may act as a key histone mark mediating crosstalk between Nt-acetylation of H4 and acetylation and methylation of histone H4 tail. However, in yeast, loss of Nt-acetylation induces H4R3me2a, but not H4K5/K8/K12ac on ribosomal DNA, even though H4S1ph was not determined[14]. It is likely that the communication between NatD-mediated Nt-acetylation of histone H4

and internal acetylation and methylation is context-dependent and gene-specific.

Identification of a reliable epigenetic biomarker and related mechanisms in lung cancer will provide new insights for diagnosis and prognosis. Previous studies demonstrated that the NatA complex or Naa10p (the catalytic subunit of the NatA complex) is associated with cancer, and is crucial for maintaining proliferation and ensuring survival of various cancer cells[40–42]. Recently, NatD was shown to have an anti-apoptotic role in colorectal cancer cells through a p53-independent mechanism[17]. Agreeing with these observations, we found that NatD has an important role in promoting cancer cell migration and invasion. Furthermore, NatD expression levels were significantly elevated in lung cancer tissues compared with adjacent normal tissues, and correlated inversely with patient survival, corroborating the view that NatD promotes lung cancer progression. Therefore, these data indicate that NatD might be a useful diagnostic and prognostic molecular marker in lung cancer.

In summary, this study demonstrates a novel link between NatD-mediated Nt-acetylation of histone H4 and lung cancer progression. We show that NatD-mediated Nt-acetylation of histone H4 antagonizes serine 1 phosphorylation of histone H4 to promote EMT of lung cancer cells through epigenetic control of Slug (Fig. 8). NatD is essential for lung cancer cells to maintain a mesenchymal phenotype and to promote invasion, thus highlighting NatD inhibitor as a potential early therapeutic intervention in lung cancer patients.

## Methods

**Cell cultures and viral infection.** H1299 cells, A549 cells, LLC, and 293T cells were purchased from the Shanghai Institute of Cell Biology, Chinese Academy of Science (Shanghai, China). These cells were maintained at 37°C in a humidified air atmosphere containing 5% carbon dioxide in DMEM with 10% FCS (Invitrogen). The human lung cancer cell lines were recently authenticated by Genetic Testing Biotechnology Corporation (Suzhou, China) using short tandem repeat (STR) profiling. No cell line used in this paper is listed in the database of commonly

misidentified cell lines maintained by the International Cell Line Authentication Committee (ICLAC). All lines were found to be negative for mycoplasma contamination.

The small interfering RNA (siRNA) target sequences for RNA interference of NatD were inserted into the *Xho*I/*Hpa*I sites in the pLL3.7 lentiviral vector according to the manufacturer's recommendations (American Type Culture Collection, USA). The oligonucleotides were:

Human NatD shRNA KD1: 5′-GATGAAGAAGGTTATGTTA-3′
Human NatD shRNA KD2: 5′-GGTTGAATGTCTCCATTGA-3′
siRNA against CK2α and negative control (NC) siRNA were synthesized by RiboBio Co. Ltd (Gaungzhou, China). The oligonucleotides were:
CK2α: 5′-GAAUUAGAUCCACGUUUCA-3′
NC: 5′-UUCUCCGAACGUGUCACGU-3′

For overexpressing Slug, human *Slug* cDNA without the 3′-UTR was cloned into the retroviral vector plasmid pLVX-IRES-mCherry at unique *Eco*RI and *Bam*HI sites. Lentivirus or retrovirus production in 293T cells and infection of H1299 cells or A549 cells were performed in accordance with standard protocols[43]. Transduced cells were selected for GFP expression by flow cytometry.

**Cell viability and invasion assays**. The in vitro viability of H1299 cells was assessed using the Cell Counting Kit-8 (CCK-8, Dojindo, Japan) according to the manufacturer's protocol. Flow cytometric analysis of apoptosis was assessed by Annexin V and PI staining using the Annexin V-APC Apoptosis Detection Kit (KeyGEN BioTECH, China) according to the manufacturer's guide. For cell cycle analysis, cells were harvested and fixed at 4 °C overnight with 70% ethanol. Cells were washed twice with PBS, and their DNA was stained using a Cell Cycle Detection Kit (KeyGEN BioTECH, China). The samples were analyzed by flow cytometry (Becton Dickinson, NJ, USA), and results were analyzed with FlowJo software according to the manufacturer's instructions. For the wound healing assay, cells were plated to confluence in a 6-well plate, and the cell monolayer was scratched using a pipette tip. Representative photos were taken using a digital camera mounted on an inverted microscope (Olympus) at indicated times. Live cell imaging was performed using the HoloMonitor M4 time-lapse cytometer (Phase Holographic Imaging, Sweden). For cell migration assays, $5 \times 10^5$ cells were seeded into the upper chamber of the transwell apparatus (Corning Costar) in serum-free medium, and medium supplemented with 15% FBS was added to the bottom chamber. After 12 h, the cells on the upper surface that did not pass through the 8-μm pore-size polycarbonate filter were removed using a moistened cotton swab; the cells migrating to the lower membrane surface were fixed in 100% methanol for 20 min, stained with 0.4% crystal violet for 20 min, and counted under a microscope (Nikon) at ×100 magnification. The invasion assay was performed as described in the migration assay, except that the upper chamber was precoated with 50 μl of a matrigel solution.

**Purification of recombinant proteins and generation of anti-Nt-acetylation antibody against Nt-ac-H4**. Human *NatD* cDNA was cloned into pGEX6p-1 vector, and expression of full-length protein was induced in *E. coli* BL21 (DE3) by IPTG. The GST-tag was removed by treatment with PreScission protease (GE Healthcare Life Sciences). The mutant NatDΔ (lacking RRKG at amino acids 147–150)[11] was constructed using a site-directed mutagenesis kit (SBS Genetech, China). The oligonucleotides used to introduce the deletion were: 5′-TTGGAAAGCAAGGTGCTGGGGAAGTTCCTC-3′ and its complementary DNA. Expression and purification of NatDΔ were as described for NatD. Human *CK2α* cDNA (amino acids 1–335)[34] was cloned into pET28a vector at unique *Sal*I and *Bam*HI sites. All clones were confirmed by DNA sequencing. Expression and purification of CK2α were performed according to the manufacturer's protocol (Takara). Nt-ac-H4 specific antibody was generated by immunization of rabbits using Nt-ac-H4 peptide (amino acids 1–14) conjugated to KLH (Keyhole limpet hemocyanin) as an antigen. Subsequently, the IgG fraction from serum was purified by GenScript, Nanjing, China (Supplementary Fig. 6).

**In vitro acetylation assays**. Purified wild-type NatD or NatDΔ was incubated with either a C-terminal biotinylated histone H4 peptide (amino acids 1–31) or histones purified from H1299 cells, plus 2 μCi $^3$H-Acetyl-CoA (Amersham) as the acetyl donor in a mixture of 20 μl acetyltransferase buffer (50 mM Tris-HCl pH 8, 100 μM EDTA, 10% Glycerol, 1 mM DTT) for 2 h at 37 °C. Half of the sample of C-terminal biotinylated histone H4 peptide (amino acids 1–31) was precipitated with streptavidin beads, washed thoroughly with PBS, and subjected to liquid scintillation counting. The other half of the C-terminal biotinylated histone H4 peptide (amino acids 1–31) and the acetylated histones were resolved on a 15% (w/v) SDS-PAGE gel, stained with Coomassie blue, dried, and subjected to autoradiography.

**Western blot analysis and protein interaction studies**. Cellular proteins were extracted by RIPA lysis buffer at high salt concentration (420 mM NaCl), and western blot analysis was performed in accordance with standard protocols[43]. Scans of enhanced chemiluminescence (ECL) films showing uncropped blots are presented in Supplementary Fig. 10. The following antibodies were used for western blotting: NatD (Abcam; ab106408, 1:1000), GAPDH (MBL International; M171-3, 1:5000), Vimentin (BD Biosciences; 550513, 1:1000), E-cadherin (BD

Biosciencs; 610181, 1:1000), N-cadherin (Abcam; ab76057, 1:1000), Slug (Abcam; ab27568, 1:500), Zeb1 (ABclonal; A5600, 1:1000), Zeb2 (Abcam; ab138222, 1:500), Twist1 (ABclonal; A7314, 1:1000), Snail (Santa Cruz Biotechnology; sc-271977, 1:500), Histone H4 (PTM Biolabs; PTM-1004, 1:2000), Histone H3(Genscript; A01502, 1:1000), H4K5ac (Millipore; CS204381, 1:1000), H4K8ac (Millipore; CS204357, 1:1000), H4K12ac (Millipore; 06-1352, 1:1000), H4R3me2a (Active Motif; 39705, 1:1000), H4R3me2s (Abcam; ab5823, 1:1000), H4S1ph (Abcam; ab14723, 1:1000), CK2α (Abcam; ab70774, 1:2000), Lamin A/C (Genscript; A01455, 1:2000), and Hsp70 (Genscript; A01236, 1:1000). Peptide pulldown assays were performed according to standard protocols[43, 44]. Briefly, we coupled strepavidin beads to 2 μg C-terminal biotin-tagged 31-mer N-terminal peptides of histone H4 and to acetylated H4 (Nt-ac-H4), as well as to C-terminal biotin-tagged 20-mer N-terminal peptides of non-acetylated histone H3. The resulting streptavidin-coupled peptides were incubated with H1299 cellular extracts prepared with high salt extraction (420 mM NaCl). We eluted specifically bound protein from stringently washed beads, separated the samples by SDS-PAGE, and visualized proteins by western blot with anti-CK2α antibody.

For MST analysis[45], purified recombinant CK2α proteins were labeled with the Monolith NT-647-NHS. Labeled proteins were used at a concentration of 100 nM in PBS pH 7.4 containing 0.05% Tween-20. The concentration of peptides of either histone H4 (aa 1–31) or Nt-acetylated histone H4 (aa 1–31) ranged from 10 nM to 500 μM. The combined solution of labeled proteins and peptides were incubated for 5 min and transferred into silicon-treated capillaries. Thermophoresis was measured for 30 s on a NanoTemper Monolith NT.115 (NanoTemper Technologies GMBH) using 60% LED power and 20% laser power. Dissociation constants were calculated by NanoTemper Analysis 1.5.41 software using the mass action equation ($K_D$ formula).

**Immunofluorescence and confocal microscopy**. For immunofluorescence assays, cells were fixed with 4% formaldehyde for 5 min at room temperature. After washing cells 3 times in PBS with 0.1% Triton X-100, cells were blocked with 3% BSA for 30 min. Cells were incubated with primary antibody (E-cadherin, N-cadherin, or CK2α) for 1 h at room temperature. Following washes with PBS 0.1% Triton X-100, cells were incubated with a secondary antibody (Alexa Fluor 555 from Cell Signaling Technology; 4431 or Vetor Laboratories, TI-2000) for 1 h at room temperature. Following washes with PBS 0.1% Triton X-100, cells were stained with DAPI (Sigma) and visualized by immunofluorescence microscopy (Nikon). Sub-cellular distribution of CK2α was analyzed by confocal scanning microscopy (Olympus FV10i). The relative intracellular distribution of CK2α in each experimental sample was calculated as the nuclear to total (cytoplasmic + nuclear) ratio by measuring the intensity of the signals in each cellular compartment with the aid of ImageJ software (NIH). Measurements were performed on more than 30 cells from three independent experiments.

**RNA isolation and quantitative RT-PCR**. Total RNA from tissue samples and cultured cells was extracted using TRIzol reagent (Invitrogen). cDNAs were synthesized with a HiScript 1st Strand cDNA Synthesis Kit (Vazyme Biotech, China). Quantitative RT-PCR was performed using a FastStart Universal SYBR Green Master (Roche) according to the manufacturer's instructions in a Rotor-Gene 6000 (Corbett Research) in a final volume of 20 μl. Cycling conditions were 94 °C for 15 s, 60 °C for 30 s, and 72 °C for 30 s. Each reaction was performed in triplicate. The primer sequences for RT-PCR are listed in Supplementary Table 2 and Table 3.

**Chromatin immunoprecipitation (ChIP) assay**. ChIP assays were performed with H1299 cells in accordance with standard protocols[43]. Normal rabbit IgG served as the control. ChIP samples were analyzed by quantitative real-time PCR using the FastStart Universal SYBR Green Master (Roche). A standard curve was prepared for each set of primers using serial titration of the input DNA. The percentage of ChIP DNA was calculated relative to the input DNA from primer-specific standard curves using the Rotor-Gene 6000 Series Software 1.7. The primer sequences for ChIP are listed in Supplementary Table 4. Antibodies used were: H4S1ph (Abcam; ab14723), H4K5ac (Millipore; CS204381), H4R3me2a (Active Motif; 39705), H4R3me2s (Abcam; ab5823), CK2α (Abcam; ab70774), H3K4me3 (Abcam; ab8580), and H3K27me3 (Abcam; ab6002).

**Clinical samples and IHC staining**. Two tissue microarray (TMA) chips containing a total of 147 pairs of lung cancer samples and matched adjacent normal tissues with follow-up data were obtained from Shanghai Biochip Co., Ltd., Shanghai, China. Fresh lung cancer tissue samples and adjacent normal tissues were derived from patients undergoing surgical procedures at the Nanjing General Hospital (Nanjing, China). All of the patients or their guardians provided written consent, and the Ethics Committee from Nanjing General Hospital approved all aspects of this study. Immunohistochemical staining was performed using paraffin-embedded sections of biopsies from lung cancer patients and controls according to standard protocols by Cell Signaling Technology. Briefly, slides were incubated with anti-NatD or anti-Slug primary antibody, followed by incubation with horseradish peroxidase-conjugated goat anti-rabbit secondary antibody. Antibody binding was visualized using a 2-Solution DAB Kit (Invitrogen). Immunohistochemical staining of NatD or Slug in the tissue was scored independently by two

pathologists blinded to the clinical data according to the semi-quantitative immunoreactivity score (IRS)[46, 47] or H score[48]. Rare discordant scores were resolved by re-review of the slide and consultation between the pathologists. Category A documented the intensity of immunostaining as 0–3 (0, negative; 1, weak; 2, moderate; 3, strong). Category B documented the percentage of immunoreactive cells as 1 (0–25%), 2 (26–50%), 3 (51–75%), and 4 (76–100%). Multiplication of category A and B resulted in an IRS ranging from 0 to 12 for each tumor or non-tumor. On the basis of the IRS score, the immunoreactivity was classified as: – (IRS 0–4); + (IRS 5–6), ++ (IRS 7–9), and +++ (10–12). NatD or Slug expression in tumor samples with IRS ≤ 6 or IRS > 6 were classified as low or high expression, respectively. For the Pearson correlation scatter plot of NatD and Slug in human lung carcinoma, the H score was calculated by adding the multiplication of the different staining intensities as category A above (0–3) with the percentage of positive cells, i.e., H score (0–300 scale) = 3 × (% at 3 +) + 2 × (% at 2+) + 1 × (% at 1+). The clinical features of the patients are listed in Supplementary Table 1. For survival analyses, patient overall survivals stratified by expression of the gene of interest, were presented as Kaplan–Meier plots and tested for significance using log-rank tests. Degree of correlation between NatD and Slug patient-expression patterns was assessed via Pearson correlation analysis.

**Animal studies**. All animal care and handling procedures were performed in accordance with the National Institutes of Health Guide for the Care and Use of Laboratory Animals, and were approved by the Institutional Review Board of Nanjing University (Nanjing, China). Female SCID mice and C57BL/6 mice (6–8 week old) were purchased from the Model Animal Research Center of Nanjing University (Nanjing, China), and were maintained under specific pathogen-free conditions at Nanjing University. The sample size was chosen with adequate power on the basis of the literature and our previous experience[46] and for each experiment it is indicated in the figure legend. Prior to carrying out the experiment, mice were randomly assigned to two different treatment groups (NatD-KD or Scr). For xenograft studies, $1 \times 10^6$ cells were resuspended in 100 µl PBS and injected into the lateral tail vein. Lung nodules and progression were monitored and quantified using the bioluminescence system (Caliper IVIS Lumina XR) or by counting under a dissecting microscope. Data were normalized to the initial post-injection signal on day 0. Mice were killed at day 28 or day 30 to collect lungs, and lung nodules in serial sections were quantified microscopically. For orthotopic lung cancer implantation assays, $5 \times 10^6$ cells were resuspended in 50 µl medium containing 10 µl Matrigel and injected into the pleural cavity of 6–8 week old female nude mice (luciferase-labeled A549 cells) or C57BL/6 mice (LLC cells). Lung nodules and progression were monitored and quantified using the bioluminescence system (Caliper IVIS Lumina XR). Mice were killed at day 14 or day 21 to collect lungs, and lung nodules in serial sections were quantified microscopically. Blinding strategy when assessing the outcome was used whenever possible.

**Statistical analysis**. Data analysis was performed with the statistical program GraphPad Prism (v.6.01, La Jolla, CA). Results were presented as mean ± s.d. unless otherwise indicated. Statistical analyses were performed using two-tailed Student's $t$-test to derive the significance of the differences between two groups. $P < 0.05$ was considered to be significant.

**Data availability**. All relevant data are available within the article and Supplementary files, or available from the authors upon request.

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

## Acknowledgements

We thank members of the Zhao and Wei laboratories for helpful discussions. This work was supported by National Natural Science Foundation of China NSFC 31470750 and 31270811 (to Q.Z.), 81421091 (to R.T.), 2014CB542300 (to C.-Y.Z.), 81472820 (to J.W.), 2015M571736 (to J.J.), 2016M590442 (to M.L.), the Fundamental Research Funds for the Central Universities 020814380073, 020814380081, National Key Scientific Instrument & Equipment Development Program of China 2012YQ03026010 (to C.L.), and SQJ Bio-technologies Limited.

## Author contributions

J.J., A.C., Y.D., M.L., Ying W., Yadong W., M.N., B.Y., T.G., X.L., Z.X., and C.M. performed experiments; C.W. and M.F. purified proteins; H.D., M.K., and C.L. performed peptide binding assays; Y.S. and J.W. provided pathology expertise; K.Z., C.-Y.Z., D.C.S.H., and C.D.A. provided ideas, reagents, and discussion; R.T., C.K.Z., J.W., and Q.Z. designed the project and wrote the manuscript.

## Additional information

**Competing interests:** The authors declare no competing financial interests.

