## [Peer Review File · Nature Communications]

Reviewers' comments:

Reviewer #1 (Remarks to the Author):

In this manuscript Ju et al. investigated the function of NatD - an acetyltransferase responsible for Nt-acetylation of the histone H4 and H2A - in lung cancer progression. They reported that NatD is upregulated in squamous carcinomas and adenocarcinomas of the lung in human patients and found that high NatD expression is associated with poor prognosis. They then depleted NatD using shRNAs in H1299 and A549 cells and showed that loss of NatD (1) suppressed migration and invasion in 2D culture and (2) attenuated tumor formation following tail vein injection. They attributed these phenotypes to loss of Slug activation and impairment of the epithelial-mesenchymal transition program. They went on to show that NatD modifies the serine residue at the N terminus of histone H4 and that this reaction antagonizes phosphorylation of this serine by CK2 α . They proposed the inhibition of H4S1 phosphorylation is responsible for activation of Slug expression.

Although the reported function of NatD in lung cancer is interesting, this study suffers from major logical leaps and the conclusion is not sufficiently supported by the experimental evidence. In order to provide experimental support of the proposed NatD→Slug→EMT cascade the following points need to be addressed. These difficulties render the manuscript unacceptable for publication in its present form and would require substantial additional experimental work in order to address the difficulties enumerated below.

1. The authors concluded that the major down stream target of NatD is Slug, yet given the profound loss of H4S1 acetylation upon NatD KD (Fig. 5e), it is highly likely that the phenotype the authors observed is caused by global changes in gene expression. In fact, besides changes of Slug expression levels, the authors also observed expression changes of other EMT markers, such as N-cad, Vim, and E-cad. Given that NatD itself can directly affect gene expression levels, the authors cannot rule out the possibility that NatD itself directly controls other genes responsible for migration and invasion and that this function is independent of its ability to regulate Slug expression.

2. In Figure 4e-f, the authors showed that Slug over-expression could rescue the migration and invasion defects of the NatD-KD cells in vitro. However, such results cannot be used to prove the NatD→Slug→EMT cascade. Rather, they only demonstrate that NatD and Slug can each activate the migration and invasion program.

3. The authors reported that NatD-KD cells failed to undergo EMT when treated with TGF- β 1 (Fig. 4a) and attributed this to downregulation of Slug in NatD-KD cells versus control cells. However, in order to understand precisely why NatD-KD cells failed to respond to TGF- β 1, the authors need to compare the expression levels of the EMT-TFs in NatD-KD cells versus control cells under TGF- β 1 treatment. It would be interesting if not critical to know whether the transcription output of TGF- β signaling is directly affected by H4S1 acetylation status.

4. To strengthen the conclusion that NatD specifically affect Slug expression but not other EMT-TFs, the authors should compare H4S1-acetylation and H4S1-phosphorylation at the promoters of yet other EMT-TFs such as Twist, Snail, Zeb1, and Zeb2. They should compare the impact of NatD loss on other histone modifications at these promoters.

Reviewer #2 (Remarks to the Author):

In the present ms, the authors study the role of NatD in lung cancer progression and describe that

N-terminal acetylation of histone H4 competes with S1 phosphorylation by CK2a and relate this finding to upregulation of Slug and tumor progression. They first describe a correlation between NatD expression in lung cancer tumors (n=147) with lymph node status and poor survival and then show that knocking down NatD prevents EMT and tumor progression in two lung cancer cells associated to Slug downregulation. Slug downregulation also occurs in lung cancer cells transfected with an inactive NatD form. In addition, Slug expression in the tumor series correlates with NatD. Mechanistically, they characterize the epigenetic status of Slug promoter regarding Nt-Ac-H4, H4S1ph, and other histone marks in control and NatD KD cells showing the inverse relation between the two first marks and the nuclear shuttling of a small fraction of CK2a in NatD KD cells. Finally, they characterize in vitro phosphorylation of a N-terminal H4 peptide by CK2a. Overall, the ms describes a novel role for N-acetylation of H4 mediated by NatD in the control of Slug expression and its potential implication in the regulation of cellular plasticity and tumor progression in lung cancer cells, findings of potential interest to the field. This is a thorough study with experiments well designed and covering a wide range of analyses from in vivo to in vitro data. However, there are two main concerns not fully addressed in the ms that decrease the scientific quality of the ms. in its present form. The first one relates to EMT characterization in the cell model systems used by the authors, and the second one related to the functional relevance of NatD, CK2a and Slug epigenetic status with Slug expression and lung tumor progression. Besides, there are several controls that are lacking as well as important clarifications are required in the ms.

The detailed points are indicated below.

Main points

1. Fig. 1e. The Kaplan-Meier plot indicates a tumor sample (n=1926) that does not correspond to the lung tumor sample analyzed by IHC (Fig. 1b,c), lymph node status (Fig. 1d) and other clinical parameters (n=147; 73 adenocarcinomas, 74 SCC) in the present study. In addition, the survival time in the plot represented in months (0-250) seems to be at odds with the median overall survival of lung cancer patients. This important information requires clarification.
2. Two shNatD RNA sequences are used to initially characterize human H1299 lung cancer cells with similar results (Fig. 2). However, only one KD cell line is used in the rest of the ms. (except in Fig. 6a). The same applies for analyses of human A549 and mouse LLC lung cancer cells (Fig. 3 and Suppl Figs. 2 and 3). The authors need to analyze both shNatD sequences in all the models used and clarify which specific shNatD is used for H1299 cells in the rest of experiments as well as for the other two cell models. Also, it is unclear if the same shNat sequences are used for human and mouse cell lines.
3. Although H1299 cells are used for most of the experiments, the in vivo tumorigenic assays are performed on A549 cells (Fig. 3) and LLC cells (Suppl Fig. 3). The authors should also test the tumorigenic effect of NatD KD in H1299 and control cells. Also, regarding lung nodules the images shown in Fig. 3b and Suppl Fig. 3d are of low quality and do not allow to clearly see the differences reported between control and KD cells. Clarification of the method used to quantify lung nodules in both experiments is also required (serial sections are required to obtain strong conclusions).
4. Characterization of the EMT phenotype is only presented at some extent for H1299 cells in Fig. 4a,c,d. However, higher density cultures (Fig. 4a) are required to more clearly show the different TGFb response reported in control and KD cells. Moreover, IF stain of E-cadherin (Fig. 4d) indicates a faint and uneven localization at cell-cell contacts in NatD KD cells arguing against acquisition of an epithelial-like phenotype. This observation requires further characterization in high density cultures and in the absence and presence of TGFb. The consequences of NatD KD in the phenotype of A549 and LLC cells also needs to be included to firmly support the proposed role of NatD in maintenance of the mesenchymal phenotype.
5. Fig. 4b. After NatD KD in HT1299 cells the authors observe a decrease in Slug mRNA while levels of the rest of analyzed EMT factors are not altered. It is quite striking that just modifying Slug expression the phenotype and subsequent properties of HT1299 are altered. Since most EMT factors are also regulated at post-transcriptional level, it is mandatory that protein levels for the rest of EMT factors will be analyzed by WB in control and KD cells as performed for Slug (Fig. 4c).

The levels of the rest of EMT factors (mRNA and protein) in A549 and LLC cells are also required.

6. The status of the rest of EMT factors and effect on the phenotype should also be analyzed in the Slug rescue experiments (Fig. 4e to g).
7. The relation of NatD with Slug expression should also be confirmed by nuclear Slug localization, at least in H1299 cells.
8. The correlation between Slug and NatD expression in the tumor samples (Fig. 4i) should be extended to survival analyses to see whether they show a similar pattern.
9. To confirm the relation between NatD mediated Nt-acetylation of H4 and Slug expression, the authors used a catalytically inactive form of NatD, NatD-delta (Fig. 5) and slightly characterize the effect on the expression of Slug and some EMT markers (i.e., vimentin) by WB in HT1299 cells. The WB shown in Fig. 5e shows that NatD-delta is overloaded as compared to control NatD, making unclear if the consequence on Slug expression is just a matter of overexpressed inactive NatD. Proper controls and further characterization of the EMT phenotype is required to firmly sustain the conclusions from these experiments.
10. Related also to the above point, WB of Fig. 6c also requires loading control of NatD and NatD-delta forms as well as detection of Slug levels in both conditions.
11. Fig. 6. The epigenetic analyses in control and NatD KD and NatD-delta cells indicates that Nt-Ac-H4 level decreases and H4S1ph level increases in Slug promoter after Nat KD (Fig. 6b) or its inactivation (Fig. 6d). However, there is no information on the functional status of the Slug promoter in those conditions. To conclusively demonstrate this key point functional promoter assays are required. The activity of Slug promoter in the different situations (control and KD cells as well as cells transfected with active and inactive NatD) needs to be analyzed. Moreover, analyses of additional epigenetic marks in the Slug promoter associated to active (H3K4me3) and inactive (H3K27me3) gene expression should be performed in relation to NatD status to additional confirm the active/inactive promoter configuration in relation to NatD.
12. The data showing the link between CK2a and H4 modification status (Fig. 7) are only correlative. The authors show that after NatD KD the remaining cytoplasmic fraction of CK2a (about 25%) is translocated to the nuclei (Fig. 7a and b) arguing that this shuttling mediates phosphorylation of H4S1 when in the absence of functional NatD. However, it is striking to understand how the nuclear shuttling of this small fraction of CK2a over the whole 75% nuclear CK2a present in control cells can promote the change in H4Ser1 phosphorylation and Slug expression level. Is this change specific of Slug promoter or could also affect other promoter regions? Does CK2a activity really impact on the activity of Slug promoter? To fully support the involvement of CK2a the authors should analyze the epigenetic status (Nat-Ac-H4 vs h4-S1ph) in the promoter region of at least some of the EMT factors, apparently not affected by NatD (Fig. 4b). In addition, the Slug promoter activity should be tested in the presence and absence of NatD+/- CK2a. As an additional control of CK2a involvement, its activity (Fig. 7c) should be tested on an N-terminal H4 peptide mutated at S1 with a non-phosphorylated residue.

Minor points

1. Fig. 2c. Amplified images of the wound areas and quantification are required to clearly show the described differences between control and NatD KD cells. The authors should also comment on the fact that in the cell tracking analysis (Fig 2d) KD cells show lower random motility than control cells, apart from decreased overall motility.
2. It is unclear why the authors include CHEK2 in the analyses of cell cycle markers (Suppl Fig. 4a) and the arguments used to involve CCND2 in migration. Please, clarify.
3. Fig. 4h. Slug detection by WB in NatD-KD+Slug cells shows a band of lower mobility than in control cells. Please, explain.
4. Page 10, lines 7 and 8 from bottom. Conclusion: "These results indicate that NatD promotes EMT probably through activation of Slug expression" are not firmly sustained from the data, please smoothen it.
5. Page 15, lines 10-13 from top: "Our findings may help to explain.....in murine brain tissue.." This sentence is too speculative and out of scope of the present study, it should be deleted.
6. Number of experiments and replicates need to be included in all figures.

Reviewer #3 (Remarks to the Author):

Ju et al present evidence that N-terminal acetyltransferase NatD acetylates the N-terminus of histone H4 which inhibits CK2a binding to the H4 N-terminus and phosphorylation of H4S1. In their model phosphorylation of H4S1 inhibits expression of slug which is critical for epithelial-mesenchymal transition (EMT). Thus, NatD promotes EMT. They also present evidence that NatD is over-expressed in human non-small cell lung cancer and that depletion of NatD reduces lung cancer incidence and slows lung cancer growth in animal models.

This paper presents very interesting and novel findings on the role of NatD in cancer and EMT, and the molecular pathway through which NatD acts. The data supporting the role of NatD in cell line transformation and animal model tumorigenesis and its over-expression in human patients is strong. The possible involvement of CK2a and H4S1ph in this process is very interesting, but the data presented to support these claims is all quite indirect and thus does not strongly support their model. There are very straight-forward experiments that could be (and should be) performed to provide more direct evidence, as described below. In addition, the design of the experiments addressing the characteristics of cells that lead to EMT (Fig. 4) are puzzling and need to be rethought or have their rationale explained better. Also, there are many more minor aspects of the manuscript that need attention. Details for all of these comments are described below:

Major comments

- 1) There is no direct evidence that CK2a is involved in the inhibition of slug expression or for its role in phosphorylation of H4S1 on the slug promoter. This leaves unsupported major parts of the molecular model that is a major conclusion presented in the paper. This could and should be addressed via siRNA depletion and inhibitors (if CK2a-specific ones are available).
- 2) Another part of the model is that the N-ac-H4 modification blocks CK2a binding to the slug promoter, but only peptide pull-down assays are presented to support this. It should be possible to perform ChIP for CK2a on the slug promoter. This is not as critical as the previous comment, but still represents an important part of their model that is not supported by direct evidence.
- 3) Fig. 4a shows that H1299 cells are epithelial in nature until treatment with TGFb, which triggers EMT unless NatD is depleted. Figs. 4b-h then explore the role of NatD in supporting the molecular characteristics that give rise to EMT. But there is no mention of TGFb in connection with these molecular experiments, so I assume they were done on cells not treated with TGFb. If TGFb is needed to induce EMT, then it would seem much more logical to test the molecular characteristics of cells treated with TGFb, or even better to compare characteristics of cells treated or untreated with TGFb.

Minor comments

- 4) The wording in the abstract does not make the model clear, as to whether H4S1ph supports or suppresses slug expression and EMT. The wording should be clarified.
- 5) Similarly, it would be very helpful to readers if a model of the entire pathway were presented in the Discussion.
- 6) Fig. 1a does not display the data in a way that allows it to be evaluated by readers. It is impossible to see most of the data points and the lines that connect them. Authors need to find a better way to present this data.
- 7) In Supplementary Table 1, the authors should explain what the p values refer to, and they should also provide a reference to the International System for Staging Lung Cancer.

8) For Fig. 1e no information is given regarding the n=1926 samples analyzed to produce this figure.

9) In Fig. 2c the figure quality is poor - it is very hard to see the cells and the migration boundaries.

10) In Fig. 3a the authors describe a reduced growth rate for the tumors after NatD depletion, but in fact, the depleted cells just have an initial lag or die-off and then grow at the same rate as the control cells.

11) The photo in Fig. 3b does not convey any information, since the nodules are not visible.

12) Fig. 3c is not useful. It is supposed to show that tumors have different characteristics after NatD depletion, but it apparently just shows a non-transformed region of the lung tissue from mice injected with the depleted cells.

13) Similarly, Supplementary Fig. 3d is not useful for the same reason, and also no quantitative data are provided to support the claim that NatD depletion reduces the number of tumor nodules in the LLC tumor model.

14) Supplementary Fig. 5a legend says that it shows matched normal and tumor samples, but there are no pairs of such matched samples in the Figure.

15) In Fig. 5a, can the authors explain why deletion of 4 amino acids causes an apparent decrease of 4-5 kDa in the molecular weight?

16) In Fig. 7e a 33uM Kd indicates an extremely weak binding interaction. It is not clear that this would be physiologically relevant.

Response to the reviewers' comments:

Reviewer 1

Q1. The authors concluded that the major downstream target of NatD is Slug, yet given the profound loss of H4S1 acetylation upon NatD KD (Fig. 5e), it is highly likely that the phenotype the authors observed is caused by global changes in gene expression. In fact, besides changes of Slug expression levels, the authors also observed expression changes of other EMT markers, such as N-cad, Vim, and E-cad. Given that NatD itself can directly affect gene expression levels, the authors cannot rule out the possibility that NatD itself directly controls other genes responsible for migration and invasion and that this function is independent of its ability to regulate Slug expression.

Response:

We thank the reviewer for the suggestion. Indeed, at the very beginning of the project, as we observed that NatD potentially regulated cell migration and invasion, we thought that NatD might directly acetylate histone H4S1, and in turn caused the global changes in the expression of some key EMT markers such as N-cad, Vim, and E-cad, as well as key transcription factors (TFs) such as Slug. However, to our surprise, when NatD was knocked down, we found that N-cad, Vim, and Slug were all downregulated whereas E-cad, a hallmark of EMT in cancer for migration and invasion, was upregulated, which contradicts NatD's normal epigenetic role (activation of transcription generally). Because Slug is a master regulator of EMT, which suppresses E-cad expression by direct binding to its promoter, we suspected that NatD regulated migration and invasion likely through manipulating Slug. Moreover, we found that the expression of EMT markers (N-cad, Vim, and E-cad), and the migratory and invasive capabilities of lung cancer cells were rescued by ectopic expression of Slug in NatD KD cells. These results suggest that the regulation of Slug expression by NatD might be a major pathway to control EMT in lung cancer cells. However, given the capacity of NatD to regulate expression of multiple genes, we cannot at this point completely rule out the possibility that other genes directly regulated by NatD might also contribute to migration and invasion independent of Slug expression. We have discussed this issue in the revised manuscript.

Q2. In Figure 4e-f, the authors showed that Slug over-expression could rescue the migration and invasion defects of the NatD-KD cells in vitro. However, such results cannot be used to prove the NatD→Slug→EMT cascade. Rather, they only

demonstrate that NatD and Slug can each activate the migration and invasion program.

Response:

To address the point raised by the reviewer, we have performed additional experiments in addition to the migration and invasion assays. These included the analyses of other key markers of EMT, including E-cadherin, N-cadherin, and Vimentin. We observed that Slug over-expression in NatD-KD cells restored the expression of N-cadherin and Vimentin, but repressed the expression of E-cadherin without affecting expression of Zeb1, Zeb2, Twist1 and Snail. This is in agreement with the response to Q1, NatD is less likely to directly regulate E-cad. Therefore, we favor the hypothesis that Slug is the key mediator bridging NatD and EMT. We have included these new data in the revised manuscript as Figure 4g-h.

Q3. The authors reported that NatD-KD cells failed to undergo EMT when treated with TGF- β 1 (Fig. 4a) and attributed this to downregulation of Slug in NatD-KD cells versus control cells. However, in order to understand precisely why NatD-KD cells failed to respond to TGF- β 1, the authors need to compare the expression levels of the EMT-TFs in NatD-KD cells versus control cells under TGF- β 1 treatment. It would be interesting if not critical to know whether the transcription output of TGF- β signaling is directly affected by H4S1 acetylation status.

Response:

As requested by the reviewer, we now have performed experiments to determine the expression levels of the EMT-TFs in NatD-KD cells versus control cells under TGF- β 1 treatment. In the presence of TGF- β 1, there was less reduction of Slug, N-cadherin, and Vimentin, and there was less induction of E-cadherin in NatD-KD cells versus control cells compared to effects under basal conditions. The expression of Snail, Twist1, Zeb1, and Zeb2 remained unchanged in NatD-KD cells versus control cells in the presence of TGF- β 1 which was similar to the effects under the basal conditions. These results have been added in the revised ms as Figure 4b-d. Based on our results, we speculate that the transcription output of TGF- β 1 signaling would be affected by H4S1 acetylation status. However, the identification of specific TGF- β 1 signaling pathway associated with H4S1 acetylation is beyond the scope of current study.

Q4. To strengthen the conclusion that NatD specifically affect Slug expression but not other EMT-TFs, the authors should compare H4S1-acetylation and H4S1-phosphorylation at the promoters of yet other EMT-TFs such as Twist, Snail, Zeb1, and Zeb2. They should compare the impact of NatD loss on other histone modifications at these promoters.

Response:

We thank the reviewer for the suggestion. As requested, we have now performed ChIP assays comparing H4S1-acetylation and H4S1-phosphorylation at the promoters of other EMT-TFs including Twist1, Snail, Zeb1, and Zeb2 in NatD-KD cells versus Scr control cells. We did not observe any changes of H4S1-phosphorylation at these promoters although enrichment of H4S1-acetylation on these promoters was decreased in NatD-KD cells compared to Scr cells. These results further support that NatD specifically affects Slug expression through inhibition of H4S1-phosphorylation. In addition, we have also analyzed other histone modifications at these promoters, such as histone H3K27me3 and H3K4me3. These new data have been added in Supplementary Figure 7a-d and in Figure 6c.

Reviewer 2

Q1. Fig. 1e. The Kaplan-Meier plot indicates a tumor sample (n=1926) that does not correspond to the lung tumor sample analyzed by IHC (Fig. 1b,c), lymph node status (Fig. 1d) and other clinical parameters (n=147; 73 adenocarcinomas, 74 SCC) in the present study. In addition, the survival time in the plot represented in months (0-250) seems to be at odds with the median overall survival of lung cancer patients. This important information requires clarification.

Response:

We thank the reviewer for this helpful comment. The tumor samples (n=1926) for Kaplan-Meier plot were originated from patient data in Kaplan-Meier database (<http://kmplot.com/analysis/index.php?p=service&cancer=lung>). These are different from the samples analyzed by IHC (Fig. 1b,c), lymph node status (Fig. 1d) and other clinical parameters (n=147; 73 adenocarcinomas, 74 SCC) which we identified in the present study. The Kaplan-Meier database for non-small-cell lung cancer patients was constructed from the combined lung cancer microarray database from GEO datasets, TCGA, or caArray with strict quality control. This database includes the patients who were either treated surgically, or received chemotherapy and/or radiotherapy. The majority of these patients (63%) were at stage 1. The 5 year median survival rate for stage 1a NSCLC is about 49% and for stage 1b is about 45%, which is consistent with the median overall survival for NatD high patients shown in Fig 1e. We have now added the related information and references in the revised manuscript (in the section of Methods and Figure legends, respectively).

Q2. Two shNatD RNA sequences are used to initially characterize human H1299 lung cancer cells with similar results (Fig. 2). However, only one KD cell line is used in the rest of the ms. (except in Fig. 6a). The same applies for analyses of human A549 and mouse LLC lung cancer cells (Fig. 3 and Suppl Figs. 2 and 3). The authors need to analyze both shNatD sequences in all the models used and clarify which specific shNatD is used for H1299 cells in the rest of experiments as well as for the other two cell models. Also, it is unclear if the same shNat sequences are used for human and mouse cell lines.

Response:

We thank the reviewer for this point. Indeed, we used two shNatD RNA sequences to initially characterize human H1299 lung cancer cells, and obtained similar results in cell migration and invasion assays (Fig.2). Thus, we used only one KD cell line (NatD-KD2, because the KD2 shRNA exhibited a bit better knockdown effect) for further investigation. As requested by the reviewer, in the revised manuscript, we used both shNatD RNA sequences to perform migration and invasion analyses in another lung cancer cell line A549. Again, we obtained the similar results as characterized in H1299 cells. We have added the results in revised manuscript as the Supplementary Figure 2a-c and Supplementary Figure 4b-c. These additional information are addressed in the text (Results section) as well as in the legends to Fig.3a and in Suppl Fig.3a.

As the sequence targeted by NatD shRNA (NatD-KD2) is identical in mouse and human, we thus applied this shRNA for both human and mouse cell lines, as well as in LLC-bearing mouse model.

Q3. Although H1299 cells are used for most of the experiments, the in vivo tumorigenic assays are performed on A549 cells (Fig. 3) and LLC cells (Suppl Fig. 3). The authors should also test the tumorigenic effect of NatD KD in H1299 and control cells. Also, regarding lung nodules the images shown in Fig. 3b and Suppl Fig. 3d are of low quality and do not allow to clearly see the differences reported between control and KD cells. Clarification of the method used to quantify lung nodules in both experiments is also required (serial sections are required to obtain strong conclusions).

Response:

We thank the reviewer for the suggestion. In fact, we initially tested the tumorigenic effect with H1299 cells in vivo. We monitored the mice using a bioluminescence system (Caliper IVIS Lumina XR) for over one month in three different batches of SCID mice. However, H1299 cells failed to form tumor nodules in the lung of SCID mice for unknown reasons. Therefore, based on the facts that NatD exerts the similar

effects in A549 and LLC cells, we performed the in vivo assays using human A549 cells and murine LLC cells.

We apologize for those low quality images which were formalin-fixed. We now provide images (before fixation) with better resolution, which clearly show the differences of lung nodules between control and KD mice (shown in revised manuscript as Fig. 3b and Suppl Fig. 3d). The similar results were observed using a bioluminescence system (as shown in Fig. 3c). As requested, we have also quantified the lung nodules by analysis of serial sections (Fig. 3b), and have addressed the additional information in revised methods section.

Q4. Characterization of the EMT phenotype is only presented at some extent for H1299 cells in Fig. 4a,c,d. However, higher density cultures (Fig. 4a) are required to more clearly show the different TGF β response reported in control and KD cells. Moreover, IF stain of E-cadherin (Fig. 4d) indicates a faint and uneven localization at cell-cell contacts in NatD KD cells arguing against acquisition of an epithelial-like phenotype. This observation requires further characterization in high density cultures and in the absence and presence of TGF β . The consequences of NatD KD in the phenotype of A549 and LLC cells also needs to be included to firmly support the proposed role of NatD in maintenance of the mesenchymal phenotype.

Response:

As requested by the reviewer, we have now used higher density cultures to investigate TGF- β responses, and obtained the similar results. Also, we have repeated the IF stain of E-cadherin, and show an even localization at cell-cell contacts in NatD KD cells. These experiments have been performed in the absence and presence of TGF- β with high density cultures. We have also performed experiments showing similar consequences of NatD KD in the phenotype of A549 and LLC cells. These data have been added in revised ms as Fig. 4a, Fig. 4d, Suppl Fig. 3f and Suppl Fig. 4d.

Q5. Fig. 4b. After NatD KD in H1299 cells the authors observe a decrease in Slug mRNA while levels of the rest of analyzed EMT factors are not altered. It is quite striking that just modifying Slug expression the phenotype and subsequent properties of HT1299 are altered. Since most EMT factors are also regulated at post-transcriptional level, it is mandatory that protein levels for the rest of EMT factors will be analyzed by WB in control and KD cells as performed for Slug (Fig. 4c). The levels of the rest of EMT factors (mRNA and protein) in A549 and LLC cells are also required.

Response:

We thank the reviewer for the suggestion. We now have performed WB for the rest of EMT factors as performed for Slug. In addition, similar experiments (both mRNA and

protein detection) were performed in A549 and LLC cells. These results are now added in revised ms as Fig. 4b-c, Suppl Fig. 3a, 3e, and Supple Fig. 4b-c.

Q6. The status of the rest of EMT factors and effect on the phenotype should also be analyzed in the Slug rescue experiments (Fig. 4e to g).

Response:

As requested, we have performed experiments to determine both mRNA and protein levels of the rest of the EMT factors in the Slug rescue experiments. These new data are shown in Fig. 4g-h in the revised ms.

Q7. The relation of NatD with Slug expression should also be confirmed by nuclear Slug localization, at least in H1299 cells.

Response:

As requested, we have performed IF staining of Slug in H1299 cells to confirm its relation with NatD expression. These new data are shown Suppl Fig. 5a in the revised ms.

Q8. The correlation between Slug and NatD expression in the tumor samples (Fig. 4i) should be extended to survival analyses to see whether they show a similar pattern.

Response:

We thank the reviewer for the suggestion. We have performed survival analyses based on expression of Slug using the Kaplan-Meier database. Indeed, Slug and NatD expression show a similar pattern with patient survival. These new data are included in Suppl Fig. 5d in the revised ms.

Q9. To confirm the relation between NatD mediated Nt-acetylation of H4 and Slug expression, the authors used a catalytically inactive form of NatD, NatD-delta (Fig. 5) and slightly characterize the effect on the expression of Slug and some EMT markers (i.e., vimentin) by WB in HT1299 cells. The WB shown in Fig. 5e shows that NatD-delta is overloaded as compared to control NatD, making unclear if the consequence on Slug expression is just a matter of overexpressed inactive NatD. Proper controls and further characterization of the EMT phenotype is required to firmly sustain the conclusions from these experiments.

Response:

We have repeated WB experiments, and found comparable amounts of NatD-delta and control NatD. We have also analyzed markers of EMT, and migration and invasion assays between NatD and NatD-delta cells to confirm the relation between NatD-mediated Nt-acetylation of H4 and Slug expression. These new data are included in new Fig. 5d-g.

Q10. Related also to the above point, WB of Fig. 6c also requires loading control of NatD and NatD-delta forms as well as detection of Slug levels in both conditions.

Response:

As requested, we have added loading controls for NatD and NatD-delta forms, and we have also performed WB for Slug expression under both conditions. These new data are included in revised Fig. 6d.

Q11. Fig. 6. The epigenetic analyses in control and NatD KD and NatD-delta cells indicates that Nt-Ac-H4 level decreases and H4S1ph level increases in Slug promoter after Nat KD (Fig. 6b) or its inactivation (Fig. 6d). However, there is no information on the functional status of the Slug promoter in those conditions. To conclusively demonstrate this key point functional promoter assays are required. The activity of Slug promoter in the different situations (control and KD cells as well as cells transfected with active and inactive NatD) needs to be analyzed. Moreover, analyses of additional epigenetic marks in the Slug promoter associated to active (H3K4me3) and inactive (H3K27me3) gene expression should be performed in relation to NatD status to additionally confirm the active/inactive promoter configuration in relation to NatD.

Response:

Since NatD and NatD-delta epigenetically regulate Slug expression through changes of histone H4S1ac and H4S1ph, it is less likely to perform the promoter reporter assay to test active and inactive NatD on Slug promoter activity in cytoplasm where functional nucleosomes (composed of histone and DNA) are not formed. Instead, we have performed additional ChIP assays using antibodies against H3K4me3 and H3K27me3 in the Slug promoter. The results of these experiments show the correlated active/inactive promoter configuration in relation to NatD. These data are included in revised Fig. 6c.

Q12. The data showing the link between CK2a and H4 modification status (Fig. 7) are only correlative. The authors show that after NatD KD the remaining cytoplasmic fraction of CK2a (about 25%) is translocated to the nuclei (Fig. 7a and b) arguing that this shuttling mediates phosphorylation of H4S1 when in the absence of functional NatD. However, it is striking to understand how the nuclear shuttling of this small fraction of CK2a over the whole 75% nuclear CK2a present in control cells can promote the change in H4Ser1 phosphorylation and Slug expression level. Is this change specific of Slug promoter or could also affect other promoter regions? Does CK2a activity really impact on the activity of Slug promoter? To fully support the involvement of CK2a the authors should analyze the epigenetic status (Nat-Ac-H4 vs

h4-S1ph) in the promoter region of at least some of the EMT factors, apparently not affected by NatD (Fig. 4b). In addition, the Slug promoter activity should be tested in the presence and absence of NatD[±] CK2 α . As an additional control of CK2 α involvement, its activity (Fig. 7c) should be tested on an N-terminal H4 peptide mutated at S1 with a non-phosphorylated residue.

Response:

We thank the reviewer for the suggestion. In order to test the specificity of CK2 α for histones around the Slug promoter, we have performed ChIP assays using antibodies against histone Nt-Ac-H4 and H4S1ph on the Slug promoter and on promoters of other EMT-TFs in NatD-KD and control cells. We found that shuttling of CK2 α in NatD-KD cells increased histone H4S1ph on the Slug promoter, but not on promoters of other tested EMT-TFs (Zeb1, Zeb2, Twist1 and Snail), although enrichment of Nt-Ac-H4 on their promoters was decreased. Thus, in NatD KD cells, CK2 α was not able to phosphorylate histone H4S1 on promoters of other EMT-TFs. The reason for this is currently unknown. In addition, we have examined Slug expression in NatD-KD and control cells with or without CK2 α expression. Under the basal condition, Slug expression was slightly increased by interfering CK2 α expression. However, in NatD KD cells, Slug expression was greatly increased by interfering with CK2 α expression, suggesting that phosphorylated histone H4S1 was required for Slug repression. When an N-terminal H4 peptide was mutated from S1 (Serine) to A (Alanine), a non-phosphorylatable residue, it did not bind CK2 α , suggesting that the required phosphorylation was serine-specific. These new data have been added in revised Fig. 7g-h, Fig. 7c and Suppl Fig. 7a-d.

Minor points

Q1. Fig. 2c. Amplified images of the wound areas and quantification are required to clearly show the described differences between control and NatD KD cells. The authors should also comment on the fact that in the cell tracking analysis (Fig 2d) KD cells show lower random motility than control cells, apart from decreased overall motility.

Response:

As requested, we have repeated the wound healing experiments and replaced the images with clearer ones. Also, we have quantified the wound healing to show the differences in cell migration. These new data are included in revised Fig. 2c. We have also discussed the fact of lower random motility in NatD KD cells than in control cells from the cell tracking analysis.

Q2. It is unclear why the authors include CHEK2 in the analyses of cell cycle markers (Suppl Fig. 4a) and the arguments used to involve CCND2 in migration. Please, clarify.

Response:

Although Checkpoint kinase 2 (CHEK2, or CHK2), a serine/threonine kinase, is essential in the cell-cycle checkpoint in response to the DNA break response, activated CHEK2 can phosphorylate multiple downstream effectors involved in regulating cell-cycle arrest, apoptosis, mitotic spindle assembly, and chromosomal stability. Thus, we surmised that CHEK2 might be a key cell cycle molecule involved in the pathogenesis of cancer. Because the current study pertains to lung cancer, we tested CHEK2 expression along with other cell cycle markers in relation to NatD. We have referred to more related references in the revised manuscript. Since NatD is not yet known to be involved in cell cycle, we wanted to verify gene expression of key molecules involved in cell cycle, including CCND2, a member of the D-type cyclins. We found that all tested cell cycle genes remained unchanged, except that CCND2 was decreased in NatD KD cells compared to control cells. These data suggested that the role of CCND2 in this context might be associated with cell migration rather than cell cycle regulation. A role for CCND2 in cell migration has previously been reported [Ladam F, et al. *Molecular Cancer Research*. **11**, 1412-1424 (2013)].

Q3. Fig. 4h. Slug detection by WB in NatD-KD+Slug cells shows a band of lower mobility than in control cells. Please, explain.

Response:

The apparent lower mobility band in WB of NatD-KD+Slug cells than control cells may be due to overloading the sample. We have repeated the experiments, and the result shows a band with a same mobility in NatD-KD+Slug cells and in control cells. These new data are included in revised Fig. 4h.

Q4. Page 10, lines 7 and 8 from bottom. Conclusion: “These results indicate that NatD promotes EMT probably through activation of Slug expression” are not firmly sustained from the data, please smoothen it.

Response:

As requested, we have revised the conclusion as: These results suggest that the ability of NatD to promote EMT likely involves activation of Slug expression.

Q5. Page 15, lines 10-13 from top: “Our findings may help to explain.....in murine brain tissue..” This sentence is too speculative and out of scope of the present study, it should be deleted.

Response:

As requested, we have deleted the sentence.

Q6. Number of experiments and replicates need to be included in all figures.

Response:

We thank the reviewer for this helpful comment. We have now included the number of experiments and replicates in all figures.

Reviewer 3

Major comments

Q1. There is no direct evidence that CK2 α is involved in the inhibition of slug expression or for its role in phosphorylation of H4S1 on the slug promoter. This leaves unsupported major parts of the molecular model that is a major conclusion presented in the paper. This could and should be addressed via siRNA depletion and inhibitors (if CK2 α -specific ones are available).

Response:

We thank the reviewer for the suggestion. We now have performed RNA interference experiments via siRNA to deplete CK2 α expression. We found that Slug expression was greatly activated when CK2 α was knocked down by siRNA, particularly in NatD-KD cells, suggesting that CK2 α is involved in the inhibition of slug expression. These data are included in new Fig. 7g-h.

Q2. Another part of the model is that the N-ac-H4 modification blocks CK2 α binding to the slug promoter, but only peptide pull-down assays are presented to support this. It should be possible to perform ChIP for CK2 α on the slug promoter. This is not as critical as the previous comment, but still represents an important part of their model that is not supported by direct evidence.

Response:

We agree with the reviewer. We now have performed ChIP analyses on Slug promoter using CK2 α antibody in NatD-KD cells compared to Scr control cells. We found that when NatD was knocked down, CK2 α was indeed significantly enriched more on the Slug promoter, which supports the peptide pull-down assays. These data are included in revised Fig. 7f.

Q3. Fig. 4a shows that H1299 cells are epithelial in nature until treatment with TGF β , which triggers EMT unless NatD is depleted. Figs. 4b-h then explore the role of NatD in supporting the molecular characteristics that give rise to EMT. But there is no

mention of TGFb in connection with these molecular experiments, so I assume they were done on cells not treated with TGFb. If TGFb is needed to induce EMT, then it would seem much more logical to test the molecular characteristics of cells treated with TGFb, or even better to compare characteristics of cells treated or untreated with TGFb.

Response:

We thank the reviewer for the suggestion. We have performed additional experiments to compare the molecular characteristics of H1299 cells treated with or without TGFβ1. These data are included in revised Figure 4b-d.

Minor comments

Q4. The wording in the abstract does not make the model clear, as to whether H4S1ph supports or suppresses slug expression and EMT. The wording should be clarified.

Response:

H4S1ph suppresses slug expression and inhibits EMT. We have clarified this in the abstract.

Q5. Similarly, it would be very helpful to readers if a model of the entire pathway were presented in the Discussion.

Response:

We thank the reviewer for the suggestion. We have incorporated a model of the entire pathway in the Discussion. The model is included in new Fig. 8.

Q6. Fig. 1a is does not display the data in a way that allows it to be evaluated by readers. It is impossible to see most of the data points and the lines that connect them. Authors need to find a better way to present this data.

Response:

We thank the reviewer for the comment. The samples in Fig. 1a are paired, and we have redrawn the dots and lines in the graph to improve readability, which should allow readers to evaluate the data.

Q7. In Supplementary Table 1, the authors should explain what the p values refer to, and they should also provide a reference to the International System for Staging Lung Cancer.

Response:

In Supplementary Table 1, p values were measured with two-sided Pearson χ^2 tests comparing the two parameters in each category, e.g., male vs. female, age > 60 vs. age < 60, etc. We have clarified this in the Supplementary Table 1. We have provided a reference for the International System for Staging Lung Cancer.

Q8. For Fig. 1e no information is given regarding the n=1926 samples analyzed to produce this figure.

Response:

For Fig. 1e, total numbers of patient samples (n=1926) included both NatD low (n=496) and NatD high (n=1430) samples. The tumor samples (n=1926) for Kaplan-Meier plot were originated from patient data in Kaplan-Meier database (<http://kmplot.com/analysis/index.php?p=service&cancer=lung>). We have now added the related information and references in the revised manuscript (in the section of Methods and Figure legends, respectively).

Q9. In Fig. 2c the figure quality is poor - it is very hard to see the cells and the migration boundaries.

Response:

Thank the reviewer for the comments. We have repeated the experiment, and now present more clearer images. Also, we have quantified the wound repair, and present the results in a histogram in revised Fig. 2c.

Q10. In Fig. 3a the authors describe a reduced growth rate for the tumors after NatD depletion, but in fact, the depleted cells just have an initial lag or die-off and then grow at the same rate as the control cells.

Response:

Since NatD has no influence on cell proliferation as shown in Suppl Fig. 1, the reduced growth rate for tumors of NatD-depleted cells is probably exerted by the attenuated migration and invasion at the beginning. Therefore, after colonization in the lung, cancer cells grew at the same rate. This has been clarified in the manuscript.

Q11. The photo in Fig. 3b does not convey any information, since the nodules are not visible.

Response:

We apologize for the low quality images. We have replaced the images using fresh instead of fixed lungs. Lung nodules are now visible, and are indicated in revised Fig. 3b.

Q12. Fig. 3c is not useful. It is supposed to show that tumors have different characteristics after NatD depletion, but it apparently just shows a non-transformed region of the lung tissue from mice injected with the depleted cells.

Response:

Thank the reviewer for the comment. We have now shown sections of H&E staining of lung nodules in revised Fig. 3b.

Q13. Similarly, Supplementary Fig. 3d is not useful for the same reason, and also no quantitative data are provided to support the claim that NatD depletion reduces the number of tumor nodules in the LLC tumor model.

Response:

We have replaced the images using fresh instead of fixed lungs to show tumors more clearly. We have revised the images accordingly, and combined the serial sections (H&E staining) with the whole lung images in revised supplementary Fig 3d. We have also quantified the tumor nodules in LLC tumor model and added the data to revised Supplementary Fig. 3d.

Q14. Supplementary Fig. 5a legend says that it shows matched normal and tumor samples, but there are no pairs of such matched samples in the Figure.

Response:

Thanks for pointing out the mistake in the legend. The matched H&E stainings are shown in Fig. 1b, not included in this figure. We have removed “matched” from the legend of Suppl Fig 5.

Q15. In Fig. 5a, can the authors explain why deletion of 4 amino acids causes an apparent decrease of 4-5 kDa in the molecular weight?

Response:

Thank the reviewer for this question. We have also noticed this difference in molecular weight. We have then confirmed the DNA sequence of the deletion-mutant expression plasmid, and found no other deletion in the construct. In addition, we have also constructed another deletion-mutant expression plasmid in a His-tagged vector, and purified the corresponding His-tagged proteins. These proteins also showed a similar decrease of 4-5 kDa in molecular weight of the deletion compared to wild-type protein after SDS-PAGE. We currently have no explanation for this difference.

Q16. In Fig. 7e a 33uM Kd indicates an extremely weak binding interaction. It is not clear that this would be physiologically relevant.

Response:

Fig. 7e shows an *in vitro* peptide pull-down assay, which might not accurately mimic the interaction of native proteins. Despite the micromolar affinity measured, the accompanying pull-down assays (Fig. 7c) robustly demonstrate the ability of CK2 α to discriminate between acetylated and non-acetylated tails. We speculate that under *in*

vivo conditions, there would be much stronger binding. This is supported by CHIP results shown in Fig. 6b, 6e, and Fig. 7f.

Reviewers' comments:

Reviewer #1 (Remarks to the Author):

The authors have performed a series of additional experiments to address the connection between NatD, Slug, H4S1 modifications and the EMT. They have also satisfactorily responded to the initial issues that were raised. This reviewer recommends acceptance of the manuscript.

Reviewer #2 (Remarks to the Author):

In the present revised version the authors have addressed most of the concerns raised on the original ms. by performing additional experiments and providing clarifications in the text an/or the rebuttal letter. The quality of the ms. has increased and most conclusions are more solidly backed by the experimental data. However, there still some aspects that the authors need to address before publication.

1. In relation to Query 11 where Slug promoter analyses were requested to functionally test the consequences of epigenetic changes, the authors argue "it is less likely to perform the promoter reporter assay to test active and inactive NatD on Slug promoter activity in cytoplasm where functional nucleosomes (composed of histone and DNA) are not formed". I disagree with the authors' response since it would invalidate all promoter assays performed in many different situations in which the action of "nuclear" transcription factors is analyzed with exogenous promoters that need to be translocated to the nucleus for functional activity. Thus, although I agree that the epigenetic changes might not be easily reproduced in the exogenous promoters, the authors should made the effort to perform this requested experimental approach.

2. Based in the xenograft studies performed by tail vein injection of human and mouse cell lines manipulated to KD NatD, as well as in in vitro invasion assays, the authors conclude "These data indicate that the role of NatD is conserved between humans and mice, and that NatD plays a critical role in promoting lung cancer cell invasiveness in vivo". (end of page 8, and page 9, two first lines). This conclusion would require orthotopic injection of the different cell lines and observation of in vivo invasion in primary tumors, while tail vein injection experiments only indicate NatD effect on metastasis generation that can occur either by favoring extravasation, initial lung seeding and/or metastatic outgrowth. Without more detailed information the conclusions on invasion in vivo should be tuned down

3. In relation to Q# 4 addressed to provide additional confirmation of the proposed role of NatD in maintenance of the mesenchymal phenotype, the authors have performed additional experiments with TGFb treatment (Fig. 4) in higher density cultures. Considering the results obtained and analyses of further markers (Suppl Fig. 4a), the authors conclude "Taken together, these data indicate that NatD is mainly required for maintaining the mesenchymal phenotype, and its downregulation inhibits EMT of lung cancer cells" (page 10, lines 11-13). A careful examination of the data of Fig 4 indicates that TGFb treatment in NatD KD cells indeed promotes a partial/intermediate EMT state rather than the cells "failed to undergo EMT" (page 9, lines 9-10), as deduced by the expression levels of epithelial and mesenchymal markers, as well as for the relative high levels of Slug expression observed in either WB, qPCR and IF analyses (Fig. 4b-d). Therefore, the conclusion on this important aspect should also be modified and tuned down.

Reviewer #4 (Remarks to the Author):

I appreciate the effort of the authors to address my concerns.

Response to the reviewers' comments:

Q1. In relation to Query 11 where Slug promoter analyses were requested to functionally test the consequences of epigenetic changes, the authors argue “it is less likely to perform the promoter reporter assay to test active and inactive NatD on Slug promoter activity in cytoplasm where functional nucleosomes (composed of histone and DNA) are not formed”. I disagree with the authors’ response since it would invalidate all promoter assays performed in many different situations in which the action of “nuclear” transcription factors is analyzed with exogenous promoters that need to be translocated to the nucleus for functional activity. Thus, although I agree that the epigenetic changes might not be easily reproduced in the exogenous promoters, the authors should made the effort to perform this requested experimental approach.

Response:

We agree with the reviewer that the experiment should be performed. Therefore, we cloned a 1155-bp Slug promoter fragment [Ref: Morita T et al., J Cell Biol. 2007, 179(5): 1027-1042] into the pGL3-basic luciferase vector (sequence confirmed by DNA sequencing). The activity of Slug promoter in the different situations (control and Knockdown cells as well as cells transfected with active wild-type and inactive NatD) were analyzed; pSV-beta-Gal plasmid expressing beta-galactosidase was used as the normalization control according to Cold Spring Harbor Protocol [Smale ST. Beta-galactosidase assay. Cold Spring Harb Protoc 2010: pdb prot5423]. As the results below show, we observed no significant changes in luciferase activity in the comparison of Scrambled control (Scr) to NatD-knockdown (NatD-KD) H1299 cells (Fig. A, left panel), or in the comparison of active wild-type NatD to inactive NatD mutant (NatD Δ) H1299 cells (Fig. B, left panel). Western blot analyses confirmed the expression levels of NatD (Fig. A&B, right panels). These results indicate that the epigenetic changes described in this manuscript were not reproduced in this transfection experiment using the exogenous promoter.

Q2. Based in the xenograft studies performed by tail vein injection of human and mouse cell lines manipulated to KD NatD, as well as in in vitro invasion assays, the authors conclude “These data indicate that the role of NatD is conserved between humans and mice, and that NatD plays a critical role in promoting lung cancer cell invasiveness in vivo”. (end of page 8, and page 9, two first lines). This conclusion would require orthotopic injection of the different cell lines and observation of in vivo invasion in primary tumors, while tail vein injection experiments only indicate NatD effect on metastasis generation that can occur either by favoring extravasation, initial lung seeding and/or metastatic outgrowth. Without more detailed information the conclusions on invasion in vivo should be tuned down.

Response:

We agree with the reviewer. As the reviewer suggested, we have performed orthotopic injection of the different cell lines (human and murine cells) and observed their invasion into primary tumors in vivo. The results were similar to those obtained in the tail vein injection experiments. We have included these new data in the revised manuscript as Supplementary Figure 9.

Q3. In relation to Q#4 addressed to provide additional confirmation of the proposed role of NatD in maintenance of the mesenchymal phenotype, the authors have performed additional experiments with TGFb treatment (Fig. 4) in higher density cultures. Considering the results obtained and analyses of further markers (Suppl Fig. 4a), the authors conclude “Taken together, these data indicate that NatD is mainly required for maintaining the mesenchymal phenotype, and its downregulation inhibits EMT of lung cancer cells” (page 10, lines 11-13). A careful examination of the data of Fig 4 indicates that TGFb treatment in NatD KD cells indeed promotes a partial/intermediate EMT state rather than the cells “failed to undergo EMT” (page 9, lines 9-10), as deduced by the expression levels of epithelial and mesenchymal markers, as well as for the relative high levels of Slug expression observed in either WB, qPCR and IF analyses (Fig. 4b-d). Therefore, the conclusion on this important aspect should also be modified and tuned down.

Response:

The reviewer makes a good point. In the revised manuscript, we have revised the conclusion as: “were largely, albeit incompletely, inhibited from undergoing EMT.”

REVIEWERS' COMMENTS:

Reviewer #2 (Remarks to the Author):

The authors have satisfactorily addressed the remainaing concernsby performing the requested experiments and modification of the text.

I am glad to support the ms for publication